# Determining the terrain characteristics related to the surface expression of subsurface water pressurization in permafrost landscapes using susceptibility modelling

Jean E. Holloway[1], Ashley C. A. Rudy[1], Scott F. Lamoureux[1], Paul M. Treitz[1]

[1]Department of Geography and Planning, Queen's University, Kingston, K7L 3N6, Canada

*Correspondence to*: Jean E. Holloway (jholl013@uottawa.ca)

**Abstract.** As the Arctic warms, deepening active layers and thaw of ice-rich permafrost can lead to various forms of degradation, including slope failures referred to as active layer detachments (ALDs) and expulsions of pressurized slurries

called mud ejections (MEs). ALDs and MEs both form from high pore-water pressures (PWPs) caused by rainfall events and rapid thawing of ice-lenses at the base of deep active layers. To predict areas that have the potential for high PWPs, we use susceptibility maps generated using a generalized additive model (GAM). As model response variables, we used ALDs and MEs, both found at the Cape Bounty Arctic Watershed Observatory, Melville Island, Canada. As explanatory variables, we used the terrain characteristics elevation, slope, distance to water, topographic position index (TPI), potential incoming solar

radiation (PISR), normalized difference vegetation index (NDVI; ME model only), and topographic wetness index (TWI). The susceptibility models demonstrate that ALDs are most probable on hill slopes with gradual to steep slopes and relatively low PISR of 1100 MJ·m$^{-2}$, whereas MEs are associated with higher elevation areas, lower slope angles and in areas 600m from water. Based on these results, this method identifies areas sensitive to high PWPs, and helps improve our understanding of geomorphic sensitivity to permafrost degradation.

**1 Introduction**

Unusually warm conditions during recent years in the Arctic have led to changes in the thermal, hydrological, and geotechnical properties of the active layer and the uppermost permafrost (Kokelj, 2002; ACIA, 2005; IPCC, 2013). Deepening active layers and thaw of ice-rich permafrost can lead to various forms of degradation, including slope failures referred to as active layer detachments (ALDs), and expulsions of pressurized slurries referred to here as mud ejections

(MEs, Holloway et al., 2016) (Figure 1). While these features are morphologically different, the processes causing their formation are similar: i.e., high PWPs caused by rainfall events and rapid thawing of ice-lenses at the base of deep active layers (Shilts, 1978; French, 2007; Lewkowicz, 2007). It has been documented in the literature that ALDs can be damaging to infrastructure (Nelson et al., 2002) and increase sediment and solute yields in surface waters (Lamoureux and Lafrenière, 2009). MEs represent a significant gap in the current literature, but are a surface expression of potentially hazardous high

PWPs.

Water and ice enrichment at the base of the active layer and near-surface permafrost has been well documented (Kokelj and Burn, 2005; Shur et al., 2005; Tarnocai, 2009; French and Shur, 2010). The ice-rich layer at the top of the permafrost is called the transient layer, as it undergoes episodic thaw during exceptionally warm years with thick active layer formation (Hinkel et al., 2001; Kokelj and Burn, 2003; Shur et al., 2005). In this way, during warm years with deep thaw the transient layer becomes part of the active layer, but during cold years it remains part of the permafrost. The ice-rich zone develops through ice segregation, or by infiltration of precipitation or melting ground ice in the active layer and subsequent refreezing at the top of the permafrost (Mackay, 1983; Hinkel et al., 2001). When the transient layer thaws during exceptionally warm years the ice melts and creates excess water at the base of the active layer. This addition of moisture from melting of ice-rich soil, as well as infiltration from late season precipitation, results in high pore-water pressures (PWP) at the base of the active layer (Zoltai, 1978; Wang et al., 2005; Yamamoto, 2014). PWP is the pressure that the water in the voids of saturated soil is under, and influences the shear strength of the soil (Mitchell, 1960; Morgenstern and Nixon, 1971; McRoberts, 1978). Similarly, PWP generation at the thawing front and from deep thaw into the transient layer causes slope instability (McRoberts and Morgenstern, 1973; Harris, 1981; Lewkowicz and Clarke, 1998; Andersland and Ladanyi, 2004). Since these pressures can lead to potentially hazardous forms of permafrost degradation and disturbance, it is important to understand how pore-water pressurization occurs across the landscape, particularly in relation to terrain variables. PWPs, ALDs, and MEs are caused by either intrinsic variables (slope angle, soil moisture, drainage patterns, solar radiation) or extrinsic variables that act as triggers (major rainfall events and increased temperatures) (Wu and Sidle, 1995; Atkinson and Massari, 1998). Although extrinsic variables are important for the formation of ALDs and MEs, all areas across the landscape experience extrinsic variables like major rainfall events and increased temperatures relatively homogeneously and do not explain sensitivity of the landscape to PWP. It appears that only certain locations which have high PWPs, ALDs, and MEs due to specific qualities of the landscape at these locations. Therefore, we are using this modeling approach which identifies the spatial distribution of intrinsic landscape factors contributing to high PWP.

To predict areas at the landscape scale that have the potential for subsurface pressurization, we use susceptibility maps generated using predictive modeling approaches (Rudy et al., 2016a). Susceptibility mapping is based on the assertion that conditions which led to geomorphologic features in the past, will also result in that same feature in the future (or present) (Varnes et al., 1984). Thus, areas identified as susceptible will have terrain characteristics similar to those in areas where this feature has already occurred. Recent landslide susceptibility studies in non-permafrost settings have begun to use nonlinear generalized additive models (GAM; Goetz et al., 2011; Niu et al., 2014; Petschko et al., 2014). This research builds upon recent permafrost disturbance susceptibility modelling that used GAMs (Rudy et al., 2016a; Rudy et al., 2016b). For our susceptibility modelling we have used both ALDs and MEs. These latter features represent the surface manifestation of ephemeral high PWPs (Holloway et al., 2016) and are particularly important because MEs may act as an indicator of potentially hazardous pressures that may lead to slope failure. Hence, the objectives of this research are to a) independently identify the modelled susceptibility regions for both ALD and ME features to identify key landscape variables contributing

to their occurrence, and b) compare the landscape position of these features to predict areas susceptible to high PWP and landforms caused by them.

## 2 Study Site

This study was undertaken at the Cape Bounty Arctic Watershed Observatory (CBAWO), on Melville Island, Nunavut, Canada (74°43'N, 109°35'W; Figure 2). CBAWO is a multi-disciplinary research station where monitoring of terrestrial and aquatic ecosystems has been ongoing since 2003. The landscape consists of rolling hills and broad valleys with relief generally <100 m. The study site is underlain by thick continuous permafrost with a seasonally thawed active layer that can reach 0.7-0.9 m by late July (Lewis et al., 2012; Rudy et al., 2013). Permafrost cores taken near an ALD at the site show ice enrichment (>50 % ice) from 60-80 cm below ground surface (Lamhonwah et al., 2017). Similarly, observations in the headwalls of ALDs show ~0.5 m of massive ice starting at ~80 cm. CBAWO is climatically a polar semi-desert, with a mean annual air temperature (based on the nearest long-term weather station at Mould Bay, NWT, 300 km west) of -17.5°C (1971 to 2000) (Environment Canada, 2014). Precipitation primarily occurs as snow, which is extensively redistributed by wind and preferentially deposited on leeward slopes and in low-lying areas. Rainfall is infrequent with total June-July precipitation averaging 33 mm over the 2003-2013 record, but high-magnitude events of 10-35 mm over 24 hours do occur (Favaro and Lamoureux, 2014). Vegetation is composed of graminoid, prostrate dwarf shrub, and forb tundra. Vegetation cover is heterogeneous and varies across the landscape reflecting soil moisture and drainage conditions across a mesotopographic gradient (Atkinson and Treitz, 2012).

The site is underlain by sandstone and siltstone bedrock, but outcrops are uncommon (Harrison, 1995). The dominant surficial materials are late Quaternary glacial and marine sediments of unknown thickness and felsenmeer (Hodgson et al., 1984). Stratigraphy of the samples taken from the active layer across the site indicates fine-grained sediments (Holloway et al., 2016).

At CBAWO, ALDs and MEs occurred extensively in late July 2007 when exceptionally high °C and hourly temperatures reaching above 20°C and two major rainfall events on 30 June (9.2 mm) and 22 July (7.2 mm)   resulted in deep active layer thaw (~1 m) (Lamoureux and Lafrenière, 2009). Active MEs were also observed in 2011 and 2012, corresponding to years with the highest mean July temperatures of 9°C and 8.8°C respectively since 2003 at CBAWO (Holloway et al., 2016) and since 1948 when measurements began at Mould Bay, NWT (Environment Canada, 2014). ALDs and MEs occur when there is rapid thaw at the base of the active layer resulting in high PWPs, and occur in similar ice-rich soil materials (Harris and Lewkowicz, 1993; Leibman, 1995; Lewkowicz and Harris, 2005; Lamoureux and Lafrenière, 2009). In the case of ALDs, these pressures lead to shear failure and downslope sliding of the active layer over the failure surface (Harris and Lewkowicz, 1993). ALDs are generally shallow, with a steep headwall or scarp and an un-vegetated slump scar often being ~1 m deep (Figure 1) (French, 2007). MEs form when high PWPs eject sediment slurries upward through pre-existing cracks or soil structures. They can occur as active (presently ejecting sediment) or inactive (dry and dormant) stratoform mounds on

level terrain, and they naturally elongate downslope when occurring on slopes (Holloway et al., 2016). Previous research completed at CBAWO has found that these features appear to occur in distinct landscape settings: i.e., ALDs are commonly found on vegetated slopes, whereas MEs occur on high-elevation, flat, less vegetated terrain (Holloway et al. 2016).

## 3 Methods

### 3.1 Data sources and processing

High-resolution (0.5m) stereo panchromatic WorldView-2 data were collected on 15 July 2012. Using these data, a high-resolution (1 m vertical resolution) digital elevation model (DEM) was derived using PCI Geomatica 10.3.2 (Collingwood, 2014).

### 3.2 Model response variables

#### 3.2.1 Active layer detachments

An inventory of ALDs produced by Rudy et al. (2013) was used in this study. The inventory of 131 ALDs was created by field mapping and through visual inspection of the WorldView-2 imagery (Figure 2). To evaluate stable landscapes, an equal number (i.e., 131) of undisturbed points were randomly selected in ArcGIS with constraints defined for distance from a water source (>10 m) and distance to an ALD (>20 m). These constraints were chosen because on average the width of channels at Cape Bounty are less than 10 m, so to ensure that randomized points were not placed in a stream a rule of >10 m was selected. Again, to ensure that randomized points were not placed within the boundary of existing ALDs a minimum distance of 20 m was selected.

#### 3.2.2 Mud ejections

The dataset used in this study for ME locations was produced by Holloway et al. (2016). Locations were determined by field mapping in June–July, 2012 and July, 2013, and include a total of 228 MEs (Figure 2). Dense clusters of MEs were removed because they resulted in bias in the analysis. Declustering was achieved by creating a 10 m buffer zone around each mapped ME in ArcGIS 10.1 and areas where buffer zones intersected were treated as one large polygon to represent the region of the cluster. One point for every 10 MEs within the polygon were randomly generated within the area as a representative point (i.e., a cluster of 25 MEs would result in 3 points). A total of 6 clusters were removed, leaving a total of 78 MEs. To evaluate control areas where there were no MEs present, 78 sites were randomly generated in ArcGIS 10.1 to correspond to the number of MEs used in the model.

### 3.3 Model explanatory variables

Variables tested in the models were elevation, slope, distance to water, topographic position index (TPI), potential incoming solar radiation (PISR), distance to water, normalized difference vegetation index (NDVI; used only for the ME model), and topographic wetness index (TWI). These variables were chosen as they all have the potential to contribute to areas having
high PWPs. Elevation (m), slope angle (°), distance to water (m), TPI, and PISR ($MJ·m^{-2}$) were derived in ArcGIS 10.1 using the DEM. Elevation is used as a proxy for marine limit in the area, with more frost-susceptible soils and ground ice content being below marine limit (~60-80 m a.s.l) (Barnett et al., 1977). Slope angle is considered an important factor in drainage and in gravitational movements like ALDs, which can occur on low gradient slopes (Niu et al., 2005; van Westen et al., 2008). Distance to downslope water sources (herein called "distance to water") is an indication of drainage and wetness of
the landscape, and water sources have the potential to erode banks and cause ALD initiation (Dai et al., 2001). Distance to water was calculated using the Euclidean Distance Tool in ArcGIS and distances were measured from a ALD or ME to a downslope hydrological vector layer (lake or river; see Figure 4 and 5). TPI compares the elevation of each cell in a DEM to the mean elevation of a specified neighborhood (50 m radius for this study) around that cell, which was used to evaluate drainage conditions for a location (Jenness, 2006; Guisan, 1999). PISR represents differences in intensity of solar radiation,
which can control local temperature, evaporation, and snowmelt and therefore soil moisture and active layer depths (van Westen et al., 2008). Total PISR was derived using the Solar Analyst program in ArcMap and calculated for the snow-free period, which is estimated to be 15 July–15 September. Insolation is partitioned into direct-beam and diffuse radiation using the mean cloud cover factor of 0.75 based on the Nav Canada Graphic Area Forecast (Hudson et al., 2001).

A normalized difference vegetation index (NDVI) was used as a proxy for vegetation cover, derived from the multispectral
Worldview-2 image acquired on 15 July 2012 (see Tucker, 1979). NDVI is a dimensionless radiometric measure that ranges from -1 (non-vegetated surfaces) to +1 (healthy, productive vegetation). NDVI was used only for the ME model, as ME location at the site was shown to be linked to vegetation cover, and vegetation cover determines patterns in soil moisture and ground ice conditions (Holloway et al., 2016). It was not used in the ALD model because original NDVI conditions are changed by ALDs as vegetation is removed (Rudy et al., 2013).

AA topographic wetness index (TWI) (Beven and Kirkby, 1979) was calculated using Whitebox Geospatial Analysis Tools (Lindsay, 2014). TWI, a proxy for soil moisture (Beven and Kirkby, 1979), is used to quantify the factors controlling hydrological processes for a given area using elevation, slope, and the upstream area contributing to any given cell. TWI provides information on where soil moisture is likely to be higher as a result of the drainage of surface and soil water. This is important as an increase in subsurface water content can lead to increased pore-water pressure which is a triggering factor for
ALDs and MEs. A FD8 flow algorithm was applied to allow water to flow into multiple neighbouring cells based on the concave or convex nature of the landscape. TWI is an indicator of the likelihood of saturated soil conditions during rain events, and represents hydrologic parameters influenced by slope morphology. Low TWI indicates drier areas, whereas high TWI (15 for our site) indicates wetter locations. While ground ice content is linked to high PWPs, it is not used as an input

variable as ground ice maps were unavailable and impractical to attain. However, cores indicate the presence of an ice-rich transient layer at the site (Lamhonwah et al., 2017).

To test for multicollinearity amongst the variables we used variance inflation factors (VIFs) and the Spearman's rank correlation coefficient ($\rho$Sp). VIFs estimate how much the variance of the regression coefficient is "inflated" because of linear dependence between the variables (Neter et al., 1996). A VIF can be calculated for each variable by deriving a linear regression of that variable on all the other variables. $\rho$Sp measures the statistical dependence between two ranked variables.

### 3.4 Generalized additive model

Modelling was performed as a case-control study with points for either ALDs or MEs as cases and randomly selected undisturbed points as controls. An equal number of undisturbed samples were randomly generated in ArcGIS to match the disturbed samples (ALDs or MEs) and resulted in an ALD dataset of 262 samples and a ME dataset of 456 samples (Table 1). The total (combined disturbed and undisturbed samples) dataset was then randomly separated into 70 % calibration and 30 % validation subsets.. This resulted in 184 points for calibration of ALDs (92 each disturbed and undisturbed), 78 for validation (39 each disturbed and undisturbed) of ALDs, 320 (160 each disturbed and undisturbed) for calibration of MEs, and 136 (68 each disturbed and undisturbed) for validation of MEs.

To model the relationship between the response variable (either ALD or ME) and the terrain variables we used a generalized additive model (GAM). GAMs are semi-parametric extensions of generalized linear models (GLMs) and provide the flexibility to represent the response's dependence on the predictor variable as either linear or nonlinear (Hastie and Tibshirani, 1990). This type of model is advantageous as nonlinear effects are known to exist in many geomorphologic studies (Goetz et al., 2011; Rudy et al., 2016a). To account for nonlinear variables, GAMs transform nonlinear predictor variables with a smoothing function. The GAMs in this study were fitted using a spline smoother with 4 degrees of freedom allowing for the detection of complex nonlinear responses. The use of GAMs in susceptibility modelling has shown strong predictive performance and has been used in susceptibility modelling with positive results (Brenning, 2008; Jia et al., 2008; Goetz et al*., 2011; Niu et al., 2014; Rudy et al, 2016; Rudy et al., 2016b). The models were developed using R (version 2.15.3, R Core Team, 2013; see Rudy et al., 2016a for details). Model selection was performed using the 'dredge' function of the R-package 'MuMIn' where the GAM was fitted through iterative evaluations of modified combinations of the terms in the global model (Barton, 2011). For both ALDs and MEs, model selection was based on two parameters, Akaike information criterion (AIC) and explained deviance. AIC measures the quality of each model in a set based on goodness-of-fit while penalizing for model complexity (Akaike, 1974). Relative importance of variables was evaluated by the change in explained deviance from the full model as variables were removed individually. If the variable is important for the model it will result in a higher explained deviance. Slope followed by PISR had the greatest explained deviance from the full ALD model, whereas it was elevation for the ME model. Explained deviance was calculated following Eq. (1):

$$Explained\ Deviance\ = \frac{(Null\ Deviance\ -\ Residual\ Deviance)}{(Null\ Deviance)}, \tag{1}$$

where "Null Deviance" is the deviance of the model with only the intercept and "Residual Deviance" is the deviance that remains unexplained after all variables have been included (Leyk and Zimmerman, 2005). Output models were ranked by AIC and all models with ΔAIC ≤ 10 were examined (Burnham and Anderson, 2004). This approach allows us to evaluate a wide range of possible models to ensure that each variable is informative and results in a model with the greatest explained deviance and lowest AIC (Hand, 1997; Hosmer and Lemeshow, 2000; Petschko et al., 2014).

### 3.5 Performance assessment

Two methods were used to assess model performance - the area under the receiver operating characteristic (AUROC) curve and a confusion matrix. The receiver operating characteristic (ROC) curve plots all possible combinations of sensitivities (i.e., percentage of correctly classified disturbance points) against the corresponding specificities (i.e., percentage of correctly classified undisturbed points) that can be achieved with a given classifier and is independent of the spatial density of disturbance (Goetz et al., 2011). Overall model performance is then determined by calculating the AUROC curve where the curve ranges from 0–1. A model that has an AUROC of 0.5 or less does not predict the occurrence of disturbance any better than chance, whereas a model with an AUROC of 1 represents a model with perfect prediction of the two classes. The quantitative-qualitative relationship between AUROC and prediction accuracy can be classified as follows: 0.9–1, excellent; 0.8–0.9, very good; 0.7–0.8, good; 0.6–0.7, average; and 0.5–0.6, poor (Yesilnacar, 2005). To assess the performance of a presence/absence classifier the agreement between predictions and actual observations can be examined using a confusion matrix. For a disturbance to be predicted as present or absent, the predicted probability will be higher or lower than a pre-assigned probability threshold. For this study, a threshold of 0.50 was selected to maximize the sensitivity to specificity ratio.

### 3.6 Permafrost disturbance susceptibility maps

For each GAM model, the results were interpolated (and extrapolated) with R packages "raster" and "rgdal" to produce a probability map that was classified into a permafrost disturbance susceptibility map. For this study, we classified our map into susceptibility zones, using the $50^{th}$, $75^{th}$, $90^{th}$, and $95^{th}$ percentiles, representing: Very Low (<50); Low (50–75); Moderate (75–90); High (90–95); and Very High (>95) susceptibility to future disturbance. This is the classification used in the current literature for susceptibility modelling, but is based on expert opinion and is not statistically tested (Dai and Lee, 2002; Ohlmacher and Davis, 2003).

## 4 Results

### 4.1 Model fit and predictive power

ALDs and MEs were accurately modeled in terms of susceptibility to future disturbance across the study area (Table 1). Terrain variables included in the final ALD model were slope, elevation, PISR, TPI, distance to water, and TWI resulting in

an explained deviance of 45 %. In the ME model distance to water, NDVI, elevation, PISR, TPI and TWI were included in the highest performing model with an explained deviance of 57 %. The ALD model calibrated with a sensitivity and specificity of 84 % and 81 %, respectively, while the ME model had a sensitivity and specificity of 91 % and 88 % (Table 1). These predictive metrics indicate that both GAM models are consistently identifying both disturbed ALDs and MEs and
undisturbed points. AUROC values were consistently high for both: 0.91 for ALDs and 0.95 for MEs (Table 1).

### 4.2 Importance of predictor variables

AUROC was used to assess the discriminatory power of individual variables outside a statistical model using single variable models (Table 2). The strongest predictors for ALDs were slope and PISR (AUROC > 70 %), followed by elevation, TPI, and distance to water (AUROC > 60 %). Although TWI was weakly related to ALD occurrence (AUROC < 60 %), it's
interactions with the other model variables led to an increase in the performance of the full model. For MEs, all variables with the exception of slope and TWI had AUROCs > 70. Differences in values between undisturbed and disturbed ALD points were checked for significance using a Wilcoxon rank sum test (for continuous variables). In the ALD model, all continuous variables with the exception of TWI and distance to water are statistically significant at the 99.9 % level whereas in the ME model all continuous variables with the exception of TWI and PISR are statistically significant at the 99.9 % level.
VIFs and $\rho$Sp were used to examine correlations between predictor variables in both models. For the ALD model, slope and PISR had the strongest correlation with $\rho$Sp equal to -0.52, all other variables had weaker correlations with $|\rho$Sp$|$<0.5. All variables in the ALD model had VIFs below two. For the ME model, elevation and distance to water had the strongest correlation with $\rho$Sp equal to 0.54, all other variables had weaker correlations with $|\rho$Sp$|$<0.5. All variables in the ME model had VIFs below two.
Bivariate plots were constructed to view the relationship between the probabilities of an ALD or a ME occurring and the terrain variables that had the largest influence on the models (Figure 3). When the slope is low the probability of an ALD occurring is also low, but as the slope increases to ~12 ° the probability peaks and then starts to decline slightly as the slope increases. The opposite is the case for MEs, where the probability is highest at low slopes and decreases with steeper slopes. For PISR, the probability of ALDs is highest when PISR of 1100 MJ·m$^{-2}$ and decreases in a logistic pattern as PISR
increases. For MEs, the probability is low when PISR ~1150–1200 MJ·m$^{-2}$ and then peaks at 1250 MJ·m$^{-2}$. For distance to water, the probability of ALDs is highest when the distance to water is small, and decreases as the distance to water becomes greater (although slightly increasing again at 500m). For MEs, the probability is lower with smaller distances to water and peaks at 600m, after which is steadily decreases as the distances get higher. For TPI, the probability of ALDs is highest with a TPI of -2.5, while for MEs is highest at a TPI of 1, indicating the ALDs occur in landscape concavities and MEs occurring
in areas that are more convex to flat. For elevation, both ALDs and MEs have low probability at low elevations, albeit the probability of ALDs is higher than that of MEs. The probability of ALDs peaks at about 50 m, where it decreases as elevation increases. For MEs, the probability is highest at 80 m and decreases as the elevation increases. When considering TWI, the probability of ALDs is high when TWI is low, and decreases as TWI increases. The probability of MEs is low

when TWI is low, peaks between 5 and 10, and then declines again. For NDVI, the probability of MEs peaks at -0.1 and declines as NDVI increases.

### 4.3 Permafrost disturbance susceptibility maps

The spatial predictions of the GAM for ALDs and MEs (Figure 4) indicate that ALDs have a high probability of occurring on moderate to steep slopes with relatively low PISR, whereas MEs are more likely to occur at high elevations with low slope angles and far from water sources (i.e., drier). For ALDs, the areas of high susceptibility on the landscape are found along river channels and on steeper upland slopes. Areas of low susceptibility are found on plateaus, far from water sources with relatively higher PISR. Although high susceptibility zones account for a small portion of the landscape, there is a greater density of disturbance at these locations (Table 3). By contrast, the areas of high ME susceptibility are found on dry, barren, uplands and plateaus.

### 4.4 Validation of ALD and ME susceptibility maps

Probability values were extracted from the independent validation datasets to validate the susceptibility maps. A threshold of 0.5 was selected to maximize the sensitivity and specificity and was used to distinguish between disturbed and undisturbed points where values > 0.50 indicate disturbance and values < 0.5 indicate undisturbed points. Again, based on the validation, the ALD and ME models performed well (Table 1). Additionally, the models were validated without a user-defined threshold resulting in an AUROC of 0.86 and 0.92 for ALDs and MEs respectively, indicating good discrimination between disturbed and undisturbed points.

### 4.5 Spatial extent of susceptibility zones

The spatial extent of moderate to high susceptibility zones was compared between the two models to identify key terrain attributes responsible for high PWPs (Figure 5, Table 4). Very little overlap exists between the models (< 1 % of the study area, Table 3). Terrain characteristics are similar between the features in low susceptibility zones, but differ in the modelled high susceptibility zones (Table 2, Table 4). ALDs are commonly found on sloped terrain while MEs tend to occur on plateaus. Slope is the main variable driving ALD initiation, while distance to water is the most important variable explaining ME formation.

## 5 Discussion

### 5.1 Landscape distribution and terrain controls over features formed by pore-water pressurization

This analysis demonstrates that the distribution of ALDs and MEs that have developed since 2007 at CBAWO are largely distinct in terms of their spatial occurrence. The terrain associated with high susceptibility to MEs include 60m elevation

sites, frequently on plateau or interfluve locations with moderate to low slope angles (Table 2). For the ME model very high susceptibility zones were far from water and had a negative NDVI value indicating more barren surfaces. High elevation sites correspond with dry, mostly barren plateau environments at the study site, which undergo deeper seasonal thaw than wetter vegetated settings (Smith et al., 2009; Woo, 2012). These areas also correspond with landscape convexities that
experience wind scour of snow which results in relatively deeper active layers (French, 2007; Woo, 2012). In warm years, this deep thaw can liberate water as ice melts from the transient layer and increase PWP. Flat or low sloping environments result in minimized hydraulic gradients and poor drainage, resulting in the accumulation of water and potentially increased PWP. TPI was a less important variable in the ME model, but the susceptibility is highest as TPI approaches 1, indicating flat areas, supporting the statements above. PISR and TWI were not significant variables in the ME model.

In contrast, ALDs are found at downslope landscape positions on low to moderate slope angles, often in areas of convergent slope drainage and topographic concavity. This pattern has been observed elsewhere (Lewkowicz and Harris, 2005; Rudy et al., 2016a; Rudy et al., 2016b). On shallow slopes water movement is decreased resulting in an overall increase in PWPs (McRoberts and Morgenstern, 1974; Leibman et al., 2014). ALDs are associated with mean elevations of 50 m, which is below the local marine limit of 60-80 m. The frost susceptibility of marine clays tend to be more ice-rich, form segregated
ice-lenses, and have low liquid limits which contributes to slope instability (Kokelj and Burn, 2005). The probability of ALDs increases with as PISR approaches 1100 MJ·m$^{-2}$. The median value for PISR at the site is 1238 MJ·m$^{-2}$, indicating that ALDs are more probable in areas that have generally lower PISR (Rudy et al., 2016b). Low PISR is commonly associated with north-facing slopes. These areas have shallower active layers with more near surface ground ice, and longer persistence of snow which leads to increased soil moisture. Saturated soil which freezes can lead to ice aggradation and increased PWP
the following summer (McRoberts and Morgenstern, 1974). TWI and distance to water were not significant in the ALD model.

While surficial materials are broadly similar across CBAWO, the landscape zonation of these two features appears to follow a slope continuum, with MEs on upslope convex areas and ALDs on mid-slope concave areas. The toposequence at our site includes: dry, exposed plateaus at c. 80m elevation, with polar-desert vegetation or bare rock; intermediate hillslopes from
40-70 m elevation, with mesic vegetation and moderate drainage; and, low hillslope areas with wet sedge meadows and river valleys from 20-40 m. It has been shown at this site that the polar-desert plateau has deeper active layers and warmer ground temperatures than mesic tundra sites, which corresponds to the presence of MEs (Holloway et al., 2016). Mesic tundra sites have colder ground temperatures due to the presence of vegetation, shallower active layers with more near surface ground ice, conditions which can lead to formation of ALDs. This toposequence corresponds with Subzone B of the Circumpolar
Arctic Vegetation Map and is comparable to other sites within this zone (Walker et al., 2002).

As stated, high PWPs can lead to various forms of slope instability including ALDs (Harris and Lewkowicz, 1993; Lewkowicz and Harris, 2005). Similarly, MEs have been documented in settings in proximity to ALDs and the occurrence of MEs is associated with deep active layer thaw and potentially with summer rainfall (Edlund, 1989; Lewkowicz, 2007; Holloway et al., 2016). While soil PWP measurements are not available to confirm pressurization in these instances, the

inferred mechanism of ME formation is diapirisation of sediment slurries from the base of the active layer caused by pore-water pressurization due to ice thaw (Holloway et al., 2016; Lewkowicz, 2007). Similar climatic conditions and active layer processes are attributed to both ALD and ME formation (Holloway et al., 2016; Lamoureux and Lafrenière, 2009). However, when reaching some threshold of PWP, the landscape response varies depending on localized terrain characteristics shown in this study, and high PWPs are expressed at the surface as ALDs or MEs.

## 5.2 Susceptibility modelling and landscape implications

Modelling results provide a means to evaluate spatial patterns of features formed by high PWPs across the entire landscape. The independently generated models show strong mutual exclusivity of the locations where ALD and ME susceptibility is high, representing ~1 % of the study area (Figures 4 and 5). Although at the landscape scale this overlap appears minimal, these areas are of particular interest since this is indicative of where we expect a transition in soil stability and hence, areas of key potential insight into fine-scale landscape controls over disturbance. Figure 6 shows a photo of one of these areas where both feature types are present on the west central part of the study area (Figure 2). These locations will be monitored in the future to determine PWP compared to areas where models suggest low susceptibility to ALD and ME, and thus lower PWP.

Considerably more of the landscape is highly susceptible (in the high and very high zones) to disturbance in terms of ALDs rather than MEs (Table 3).Terrain does not vary as much between MEs and ALDs in the low/moderate susceptibility zones, whereas it varies substantially in the very high susceptibility zone (Table 2). This landscape zonation of ME and ALD activity is consistent with field observations at CBAWO, and constitutes the first recognition of a broader landscape pattern of soil pore-water pressurization landforms. While results indicate that many areas of the CBAWO landscape appear to be less susceptible to excess pressurization processes as expressed by MEs and ALDs, we can interpret these results to reflect controls over how pressurization affects landscapes and the locations and terrain types most susceptible. Further, the mutual exclusivity of these modelled high susceptibility areas over space suggests differential responses to pressurization across different terrain factors.

The distinct zonation of ME in upland areas with low slopes indicates that soil water pressurization results in artesian fluid release at the surface in years exhibiting warm conditions and deep active layer thaw (Holloway et al., 2016). MEs also occur coincident with conditions that result in ALD formation (Figure 6) and have been observed in cases where ALD formation was initiated but ultimately stabilized (i.e. soil cracking at surface where the active layer was failing but stabilized, with mud ejections surrounding the area). These observations suggest that MEs, while clearly reflecting evidence for subsurface soil water pressurization (i.e., Lewkowicz, 2007), also likely play a stabilization role through pressure release to the surface (Urciuoli and Pirone, 2013). Collectively, these observations strongly suggest that MEs result from soil water depressurization where active layer failure and movement is not possible or necessary. By contrast, ALDs are associated with sufficient pressurization to induce slope fracturing and downslope movement. MEs appear to be caused by fluid release to the surface that reduced subsurface pressures, essentially acting as a fluid pressure release mechanism. While this process

is observed in permafrost settings where manual drainage stabilizes slopes (Andersland and Ladanyi, 2004), and in other geotechnical settings where drainage stabilization is used for landslide mitigation (Urciuoli and Pirone, 2013), it is not clear if the presence of a ME will stabilize the soil surface and prevent soil fracturing or any slope movement. Given that that ME zones occur where ALDs are not generally observed, we interpret the ME susceptibility zone to represent areas conducive to

pore-water pressurization but where slope disturbance is inhibited, primarily by low slope angles.  However, at this time it is unclear the extent to which MEs represent a soil stabilization mechanism.

The combined extent of modelled high susceptibility areas provides a key set of terrain conditions that appear to be associated with soil water pressurization during warm summers at CBAWO.  ALDs appear in many permafrost landscapes (French, 2007; Lewkowicz, 2007; Harris, 2005; McRoberts and Morgenstern, 1974; Rutter et al., 1973; Morgenstern and

Nixon, 1971), while MEs have had less attention.  The literature and our observations indicate that MEs occur on Ellesmere (Lewkowicz, 2007), Banks and Melville Islands in the Canadian High Arctic, and further south in Nunavut and the Northwest Territories in very different vegetation and terrain conditions (Shilts, 1978). Despite these observations, the broader distribution of MEs remains poorly understood, and the terrain conditions that are associated with their occurrence at CBAWO may not apply to other settings. Hence, while MEs are indicative of subsurface fluid pressurization in permafrost

settings, the function they play in dissipating potential slope instability requires further investigation, particularly with field observations to determine the localized processes of water accumulation within the subsurface.

**6 Conclusion**

Independent GAM models accurately captured the terrain controls that appear to affect the distribution of ALDs and MEs on the landscape.  The susceptibility models demonstrate that ALDs are most probable on hillslopes with gradual to steep

concave slopes and relatively low PISR, whereas MEs are associated with higher elevation areas, low angled convex slopes and in areas relatively far from water (drier).  Both features are thought to be formed when high soil PWPs are generated and dissipated, resulting in slope failure or liquefied sediment ejecting to the soil surface. This analysis reveals that these features are found in proximity, but in largely discrete areas at CBAWO.  Hence, the joint zones of susceptibility appear to be sensitive to development of high PWPs and point to a possible larger scale landscape pattern to active layer response to

climate warming.  While further research and field measurements is necessary to elucidate these patterns, these results provide new information on landforms related to high soil PWP and improves our understanding of geomorphic sensitivity to permafrost degradation and related geohazards in Arctic landscapes.

**Acknowledgements**

Funding for this research was provided by NSERC Discovery Frontiers ADAPT, NSERC Strategic and ArcticNet programs. Logistical funding was provided by Polar Continental Shelf Program, Natural Resources Canada.  Field support for JH and AR was provided by the Northern Scientific Training Program, Canadian Polar Commission.

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

**Table 1: Model performance metrics for training (70% samples) and testing (30% samples).Sensitivity and specificity refer to the ability of the respective model to accurately identify the presence and absence of disturbance. AUROC is a measure of overall model fit without the addition of a user specified threshold.**

| Model | Number of Samples | Sensitivity (%) | Specificity (%) | AUROC |
|---|---|---|---|---|
| ALD (Calibration) | 184 | 84 | 81 | 91 |
| ALD (Validation) | 78 | 85 | 72 | 86 |
| ME (Calibration) | 320 | 91 | 88 | 95 |
| ME (Validation) | 136 | 82 | 88 | 92 |

**Table 2: Mean and standard deviation (in brackets) values for terrain variables in each susceptibility zone for ALDs and MEs and performance metrics for ALD and ME single variable models.**

| | Very Low ALD/ME | Low ALD/ME | Moderate ALD/ME | High ALD/ME | Very High ALD/ME | AUROC ALD/ME |
|---|---|---|---|---|---|---|
| Slope (°) | 4(3)/6(6) | 6(4)/5(4) | 8(4)/6(5) | 1(5)1/6(6) | 17(9)/8(9) | 80/67 |
| Elevation (m) | 67(31)/63(31) | 56(25)/67(20) | 53(25)/67(16) | 50(26)/66(15) | 48(26)/63(14) | 71/79 |
| PISR (MJm$^{-2}$) | 1241(22)/1232(37) | 1231(27)/1237(20) | 1223(34)/1238(19) | 1214(41)/1239(18) | 1172(77)/1240(17) | 70/74 |
| TPI | 0.4(1.4)/0.01(1.7) | -0.3(1)/0.3(1.3) | -0.6(1.4)/0.3(1.6) | -0.9(1.8)/0.1(1.8) | -1.73(3.2)/-0.5(2.6) | 68/74 |
| TWI | 6.5(20.6)/6.2(15.7) | 5.8(2.2)/6.2(27.6) | 5.6(2.2)/6.1(2.6) | 5.5(2.2)/6.2(2.8) | 5.3(2.2)/6.5(3.4) | 57/61 |
| Distance to water (m) | 308(192)/294(202) | 287(239)/413(276) | 310(258)/627(202) | 339(277)/652(209) | 346(283)/650(279) | 66/83 |
| NDVI | --/0.18(0.17) | --/0.04(0.12) | --/-0.02(0.15) | --/-0.05(0.17) | --/-0.09(0.21) | --/80 |

**Table 3: Area and disturbance density of susceptibility zones**

| Susceptibility Zone (Probability Range) ALD/ME | Area (km$^2$) ALD/ME | Disturbance Density (# per km$^2$) ALD/ME |
|---|---|---|
| Very low (0-0.50)/(0-0.54) | 42/52.1 | 0.9/0.7 |
| Low (0.50-0.83)/(0.54-0.90) | 8.2/6.0 | 3.8/13 |
| Moderate (0.83-0.97)/(0.90-0.95) | 4.8/0.5 | 5/150 |
| High (0.97-0.99)/(0.95-0.96) | 1.9/0.1 | 8.4/180 |
| Very High (0.99-1)/(0.96-1) | 3.2/0.4 | 6.6/50 |

| Susceptibility Zone (Probability Range) ALD/ME | Area (km$^2$) ALD/ME | Disturbance Density (# per km$^2$) ALD/ME |
|---|---|---|

**Table 4: Terrain values for areas of overlap between ALD and ME models. See units for each terrain variable in Table 2.**

| Susceptibility Class | Terrain Variable | Minimum Value | Maximum Value | Mean | Standard Deviation |
|---|---|---|---|---|---|
| Very Low/Low | Elevation | -0.37 | 160.31 | 67.06 | 32.52 |
| Moderate | Elevation | 0.66 | 122.51 | 64.21 | 17.15 |
| High/Very High | Elevation | 1.44 | 108.93 | 40.40 | 21.81 |
| Very Low/Low | Distance to Water | 0.00 | 1126.16 | 297.97 | 178.95 |
| Moderate | Distance to Water | 0.00 | 1282.12 | 432.98 | 294.51 |
| High/Very High | Distance to Water | 0.00 | 1203.88 | 391.66 | 342.92 |
| Very Low/Low | NDVI | -1.00 | 0.88 | 0.17 | 0.17 |
| Moderate | NDVI | -0.82 | 0.55 | 0.07 | 0.13 |
| High/Very High | NDVI | -0.82 | 1.34 | 0.02 | 0.26 |
| Very Low/Low | PISR | 689.28 | 1346.04 | 1240.75 | 22.84 |
| Moderate | PISR | 1055.43 | 1305.06 | 1230.95 | 21.03 |
| High/Very High | PISR | 824.33 | 1314.66 | 1214.34 | 44.83 |
| Very Low/Low | Slope | 0.00 | 67.10 | 3.95 | 3.43 |
| Moderate | Slope | 0.00 | 28.37 | 5.60 | 2.97 |
| High/Very High | Slope | 0.08 | 71.92 | 16.57 | 11.89 |
| Very Low/Low | TPI | -6.32 | 40.31 | 0.37 | 1.42 |
| Moderate | TPI | -7.85 | 4.17 | 0.06 | 0.96 |
| High/Very High | TPI | -19.19 | 3.74 | -6.81 | 3.35 |
| Very Low/Low | TWI | 0.29 | 32767.00 | 6.45 | 21.73 |
| Moderate | TWI | 1.92 | 26.08 | 5.93 | 1.97 |
| High/Very High | TWI | 1.07 | 25.98 | 6.59 | 3.08 |

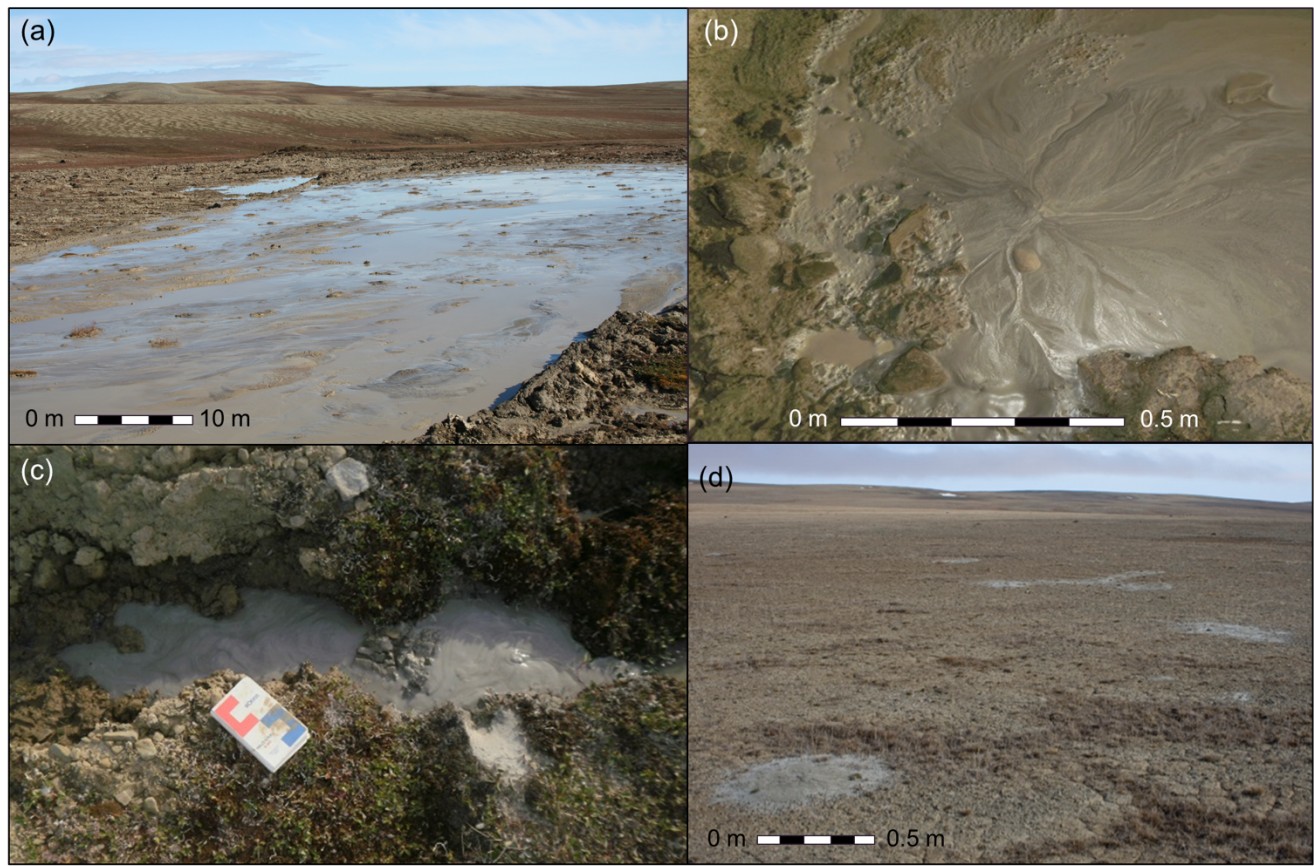

Figure 1: (a) Active layer detachment at Cape Bounty Arctic Watershed Observatory (CBAWO), Melville Island, NU, 28 July 2007. Clay slurry is evident along scar track immediately post disturbance. (b) Active mud ejection occurring on a plateau, 13 July 2012. (c) Clay slurry pooling in a crack at the headwall of a recently initiated active layer detachment, 16 July 2012. (d) Field of inactive MEs on a plateau, 18 June 2012.

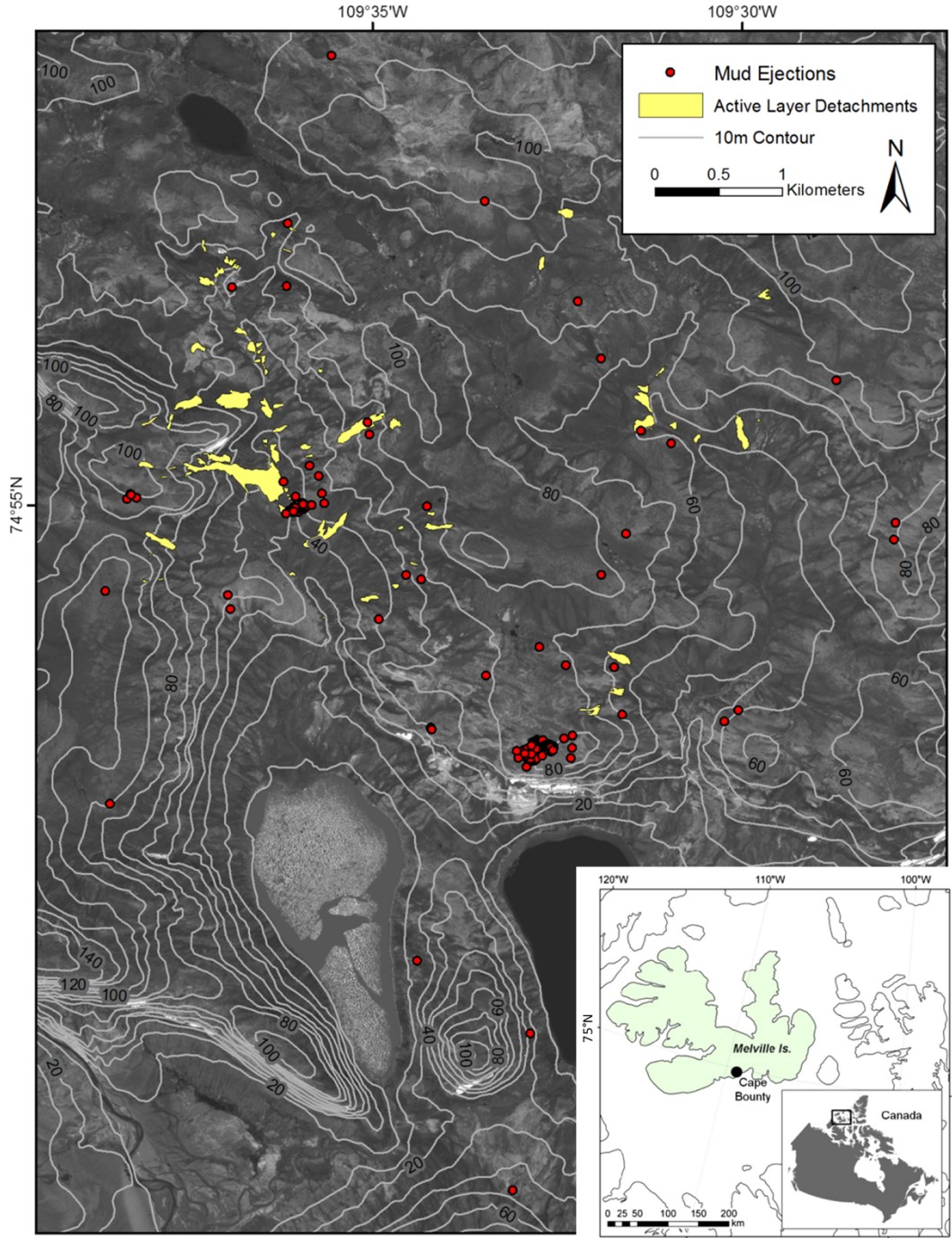

**Figure 2: Map of the study area showing locations of ALDs (Rudy et al., 2013) and MEs (Holloway et al., 2016) at CBAWO. The contour interval is 10 m and was derived from a 1 m digital elevation model (DEM) obtained from stereo GEOEYE imagery acquired in 2012 (Collingwood, 2014). The inset map indicates the site location in the Canadian High Arctic.**

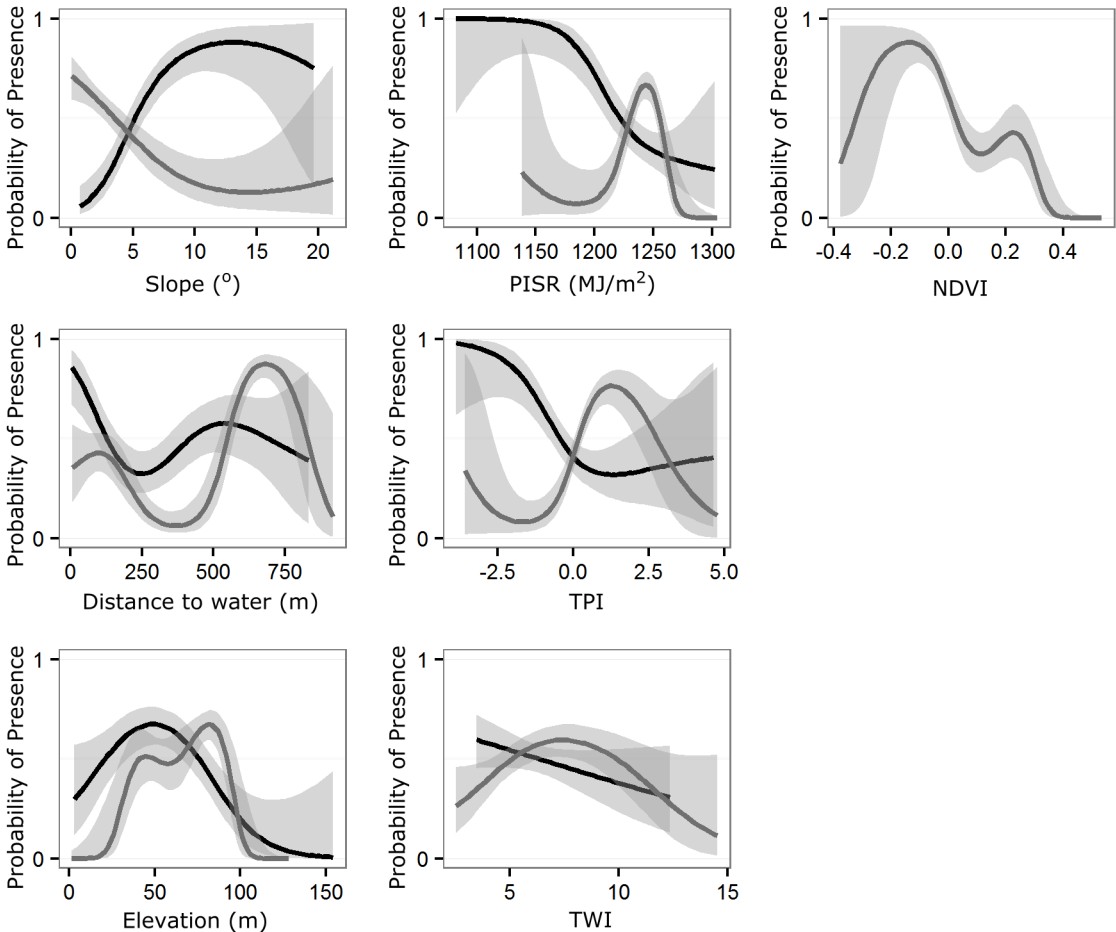

**Figure 3: Bivariate plots of the probability of the presence of ALDs (black line) and MEs (grey line) in relations to terrain variables. The grey shading illustrates the 95% confidence bands for each plot.**

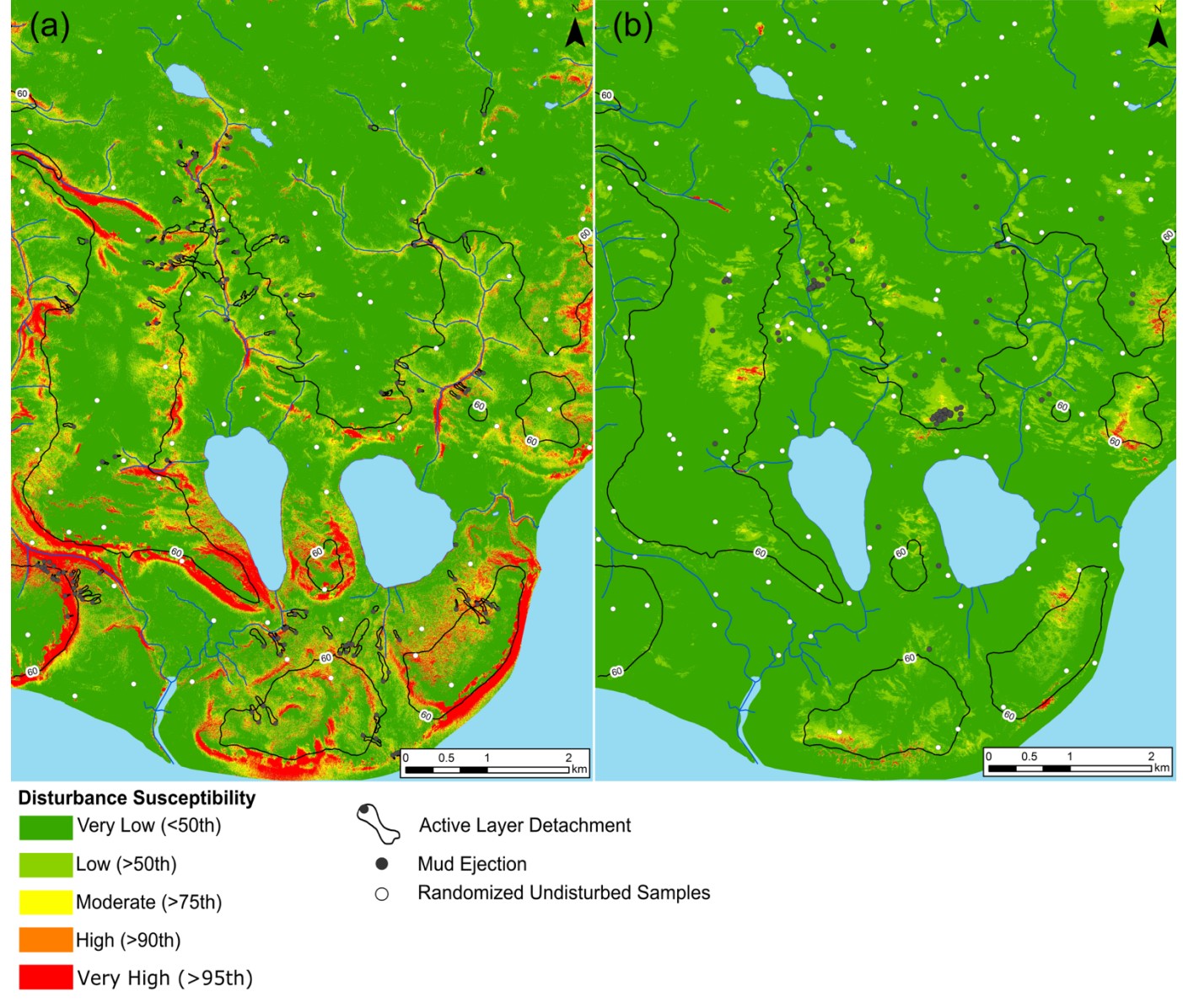

**Disturbance Susceptibility**

- Very Low (<50th)
- Low (>50th)
- Moderate (>75th)
- High (>90th)
- Very High (>95th)

- Active Layer Detachment
- Mud Ejection
- Randomized Undisturbed Samples

**Figure 4: ALD (a) and ME (b) susceptibility maps for CBAWO showing areas of very low to very high susceptibility to disturbance. Marine limit is denoted by the 60 m contour line, and the random points for undisturbed locations are indicated.**

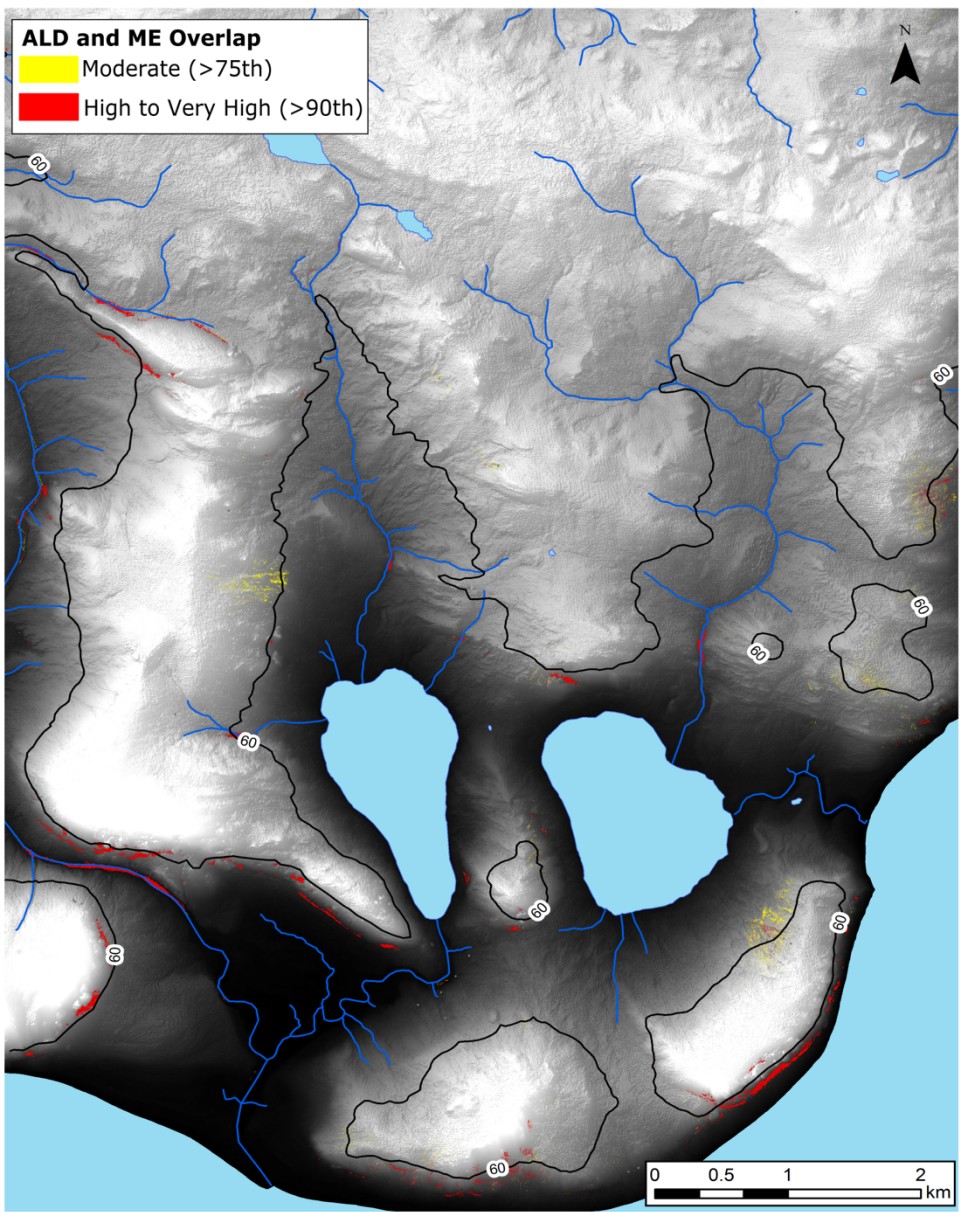

**Figure 5: Areas where the ALD and ME maps at CBAWO have overlapping susceptibilities (superimposed over the 1 m shaded relief DEM). The marine limit is denoted by the 60 m contour line.**

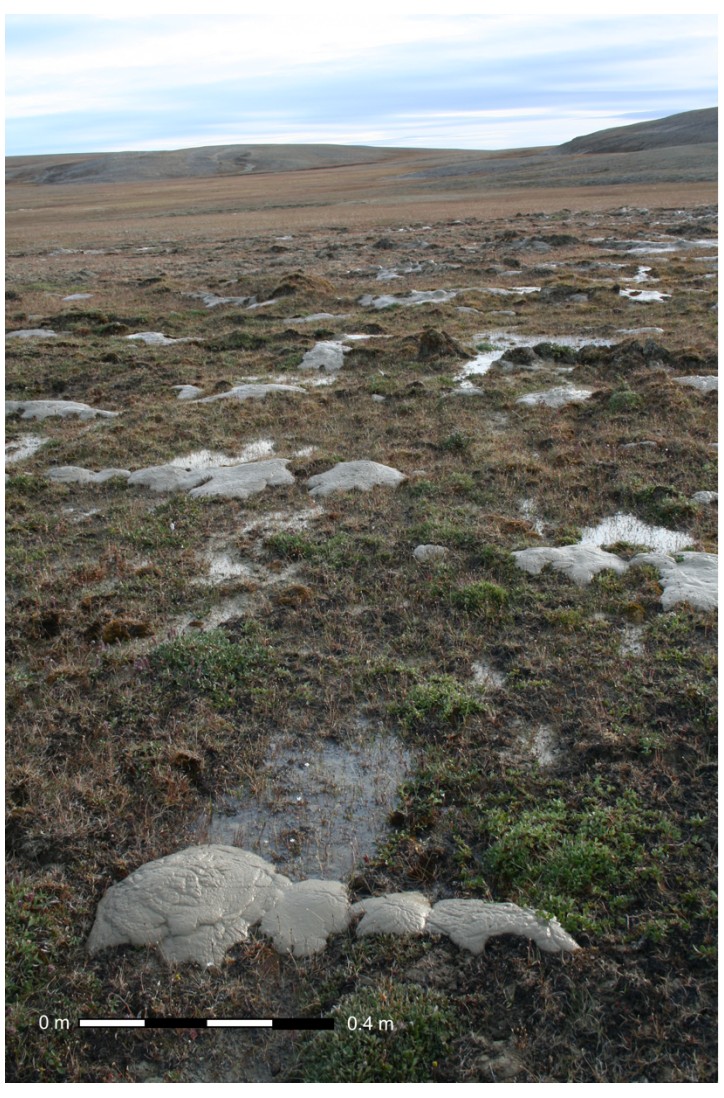

**Figure 6: MEs adjacent to an ALD at CBAWO on 28 July 2007. The photo was taken at the edge of the ALD looking out towards the MEs and the adjacent terrain.**

