# Peer review of "Determining the terrain characteristics related to the surface expression of subsurface water pressurization in permafrost landscapes using susceptibility modelling"

_The Cryosphere, 2016_

## Referee Comment (RC1) · Anonymous Referee #1 · 1 Nov 2016

**Title: Determining the terrain characteristics related to the surface expression of subsurface water pressurization in permafrost landscapes using susceptibility modelling**

**Authors: Jean E. Holloway et al.**

**General Comments:**

This paper employs GIS to define the terrain susceptible to active layer detachment slides and mud ejection features at a High Arctic site. The paper provides a suitable GIS tool to locate these features at Cape Bounty, an approach which may be applicable for similar terrain elsewhere. The paper is quite technical, with lots of jargon so I thought that it might be better suited for a GIS specific journal or PPP-*Permafrost and Periglacial Processes* where readers might be more knowledgeable about pore water pressure phenomena. If it is to be published in *Cryosphere*, more effort could be made to relate the research to other cryospheric types/regions where pore water pressure occurs to elucidate processes across diverse icy bodies. Clarification about distance to water, potential incoming solar radiation in the model is required. It was not clear from the paper whether one is concerned with upslope water or downslope water, or water sources in both directions. Also, what defines water (puddle, stream, lake or pond)? It was also not clear the time frame that PISR was calculated for (1 month, 2 months, 2 weeks)?

The study's introduction indicated that rainfall was an important factor in triggering pore water pressure but there was little information about this variable in the study. I think that more effort could also be made to get information about ground ice conditions at the site. I believe that there is a GSC report for this area, from the mid-70's which might report some of this information. Another look at field photos, or data from other studies in this location might provide clearer indication of ice content.

**Specific Comments:**

1) In the abstract you indicate distance to water but is that distance to water upslope of the feature or downslope or both? It is also not clear in the rest of the text.

2) In your abstract perhaps indicate that the GAM model is a GIS-type model.

3) Be more specific in your abstract about PISR, instead of "relatively low PISR", perhaps put a value in. Is PISR calculated for the whole summer, a few weeks? It is also not clear in your paper. Did you measure solar radiation directly at your study site? If so, how do these values compare with PISR.

4) In your abstract, perhaps put…Based on these results, this GIS method identifies….

5) At the beginning of the abstract, you indicate that late season precipitation is important for these features to develop but you don't use precipitation as an explanatory variable. In fact, I don't see any information about precipitation or late-season precipitation in the paper.

6) Again in your abstract, can you be more definite about distance…avoid saying….areas relatively far from water. Again is that upslope or downslope.

7) Page 2, Line 20-21. Perhaps cut down on the number of references to GAM.

8) Lines 27-28. Can you cut down on the references?

10) Page 3 Line 10. Put <10 m. There are some other places where you need to leave spaces…see line 14, 27, etc.

11) Line 13. Since you are concerned with the spring/summer period for slope failure, besides the mean annual temperature add information about the spring/summer temperature and also add information about these infrequent, high magnitude precipitation events.

12) Line 25. Provide information on the summer temperature in 2007 and heavy rainfall.

13) Line 27. Again put the temperatures in for 2011 and 2012, and maybe indicate how these temperatures compare with other areas in the High Arctic, and what other scientists were observing (glacial ice loss, sea ice). This will help put your work in context of other cryospheric phenomena.

14) Page 4, Line 24. Why did you select >10 m for distance to a water source and again was that upslope or downslope. Also, why did you select a distance to an ALD of > 20 m? Can you plot those randomly ArcGIS points in your map?

15) Page 5, Line 8. Can you plot the randomly generated control points for MEs (78).

16) Line 15. Do you have a reference to add to after….they all have the potential to contribute to areas having high PWPs?

17) Line 20. Again, is it distance upslope or downslope? Be specific, in terms of water, is it water table or a creek or a stream or a lake, pond. How do you define water? Many hillslope creeks in the High Arctic dry up after snowmelt, or are intermittent. Do you still estimate distance to them?

18) Line 23. Are you able to compare PISR with measured incoming radiation at a level site to see how they compare over a summer season? If you had a cloudy, rainy season then radiation across the slopes/plateau might not have been critical.

 19) Lines 7-8. Can you say more about the TWI index? How does this compare to the new paradigm of 'spill and fill', which is perhaps a better theory of how water moves in arctic environments (Woo, 2012).

20) Lines 10-11. Are you sure that you don't have any information about ground ice content. There must be some geology maps of this area which give an indication of ice content. During your fieldwork, did you not dig a hole in these different landscapes to examine where the ice and moisture were accumulating? Perhaps, look at some of your pictures, particularly, active layer detachment slides. The headwall scarps might give you an indication of where the ice rich depths occur.

21) Line 13. Do you really need $\rho$ in front of Sp? Do you have a reference for VIFs?

22) Page 8. Line 14. Do you have a reference for a confusion matrix?

23) Lines 20-23. Is this a standard framework for susceptibility/sensitivity? Should you add a reference here?

22) Line 3. In terms of PISR, how does 1100 MJ/m$^2$ compare with what is generally measured during a summer season, and what is the time frame for the PISR estimate (i.e. is this over 30, 60 or 90 days). Do you start your calculations in late August, since you said these features often occur then?

23) Line 6. Again are you referring to upslope distance or downslope distance? I would think that upslope distance to water would be more important than downslope.

24) Line 16. Indicate the amount of rain which fell late July, also indicate the depth of ground thaw.

25) Line 26. What kind of soil structure did you have which allowed these slurries to occur?

---

## Referee Comment (RC2) · Anonymous Referee #2 · 19 Jan 2017

The paper uses a general additive model and terrain characteristics derived from remote sensing to map susceptibility of permafrost disturbances (active layer detachment and mud ejection). The GIS-based analysis was successful at identifying important terrain controls at the study site, and the approach seems to have potential for application at other sites. The results are interesting and well executed and the topic is of interest to readers of The Cryosphere, but I'm not convinced The Cryosphere is the most appropriate journal. The paper is quite technical and might be appropriate for a remote sensing journal or for Permafrost and Periglacial Processes, which has a geomorphology focus. An indicator here is that not a single Cryosphere paper was cited. This

paper could be made more relevant to The Cryosphere by expanding the discussion to explore consequences for other sites, and by discussing in more depth the physical reasons for the observed explanatory power of the various terrain characteristics.

Specific comments

The title, abstract, and beginning of the paper focuses on pore water pressure, but the effect of interest is disturbance. High pore-water pressure is not observable directly, and it's possible to have high pore-water pressure without an ALD or ME. The title and the introduction should be revised to better reflect the topic of the paper - susceptibility to disturbance, not pore-water pressure.

Some of the observed relationships between the terrain variables make sense physically and some are counterintuitive. For example, why would ALD be more likely in areas of low PISR? Why would ME's be more likely in drier locations and higher elevations? Physical reasons for all the observed relationships and especially the counterintuitive ones should be explained to convince the reader that those relationships are real and not spurious correlations.

The probability of observing an ALD approaches 100% for low PISR. This is clearly site-specific and raises concerns about the transferability of the results. Please explain.

Rainfall is likely to be an important controlling variable. This needs to be discussed, since it is not addressed.

Pg 4 Line 24: what's the basis for the constraints >10 m from water source and > 20m from an ALD?

Pg 5 Line 4-7: The description of the declustering process is difficult to follow and should be explained more clearly. As I understand it, because closely located features carry redundant information, spatial clusters of features are replaced by representative features.

Pg 5. Line 4-7. The declustering algorithm seems arbitrary. Are the results sensitive to

how that is done?

Pg 6. Line 21. How were the features partitioned between the calibration and validation subsets? Random?

Pg 26. Line 1. What is an “explained deviance”?

Final sentence: The phrase “incentive and potential to move towards. . .” makes for a weak conclusion. Is it not possible to say something more definitive?

---

## Referee Comment (RC3) · Anonymous Referee #3 · 22 Jan 2017

Determining the terrain characteristics related to the surface expression of subsurface water pressurization in permafrost landscapes using susceptibility modelling

Authors: Jean E. Holloway, Ashley C.A. Rudy, Scott F. Lamoureux, Paul Treitz

General comments

In their paper, the authors evaluate the susceptibility of High-Arctic permafrost terrain to disturbances (ALD and ME) related to high pore water pressure. To do so, they used a GIS-based approach, statistics, and field validation, and made the demonstration that such an approach is exportable to other sites. The results indicate that terrain

characteristics of ALDs and MEs differed in the modelled high susceptibility zones, whereas they were similar in low susceptibility zones. They have shown that slope was the main variable driving ALD initiation and distance to water was the most important variable explaining ME formation.

Although this paper makes an interesting contribution to permafrost landscape hazards and permafrost landscape dynamics studies, I think it would be better suited for a GIS-dedicated journal or a hazards-dedicated journal. Indeed, my impression is that the cryospheric components in this article are not developed sufficiently to justify a publication in Cryosphere. If the editor decides differently, then the authors should develop a section on ground ice and particularly clarify the concept of 'transient layer' and how it applies at the landscape scale, how to model it and how to incorporate it in their GIS-based approach. A point should be made about the distribution of ground ice in a given watershed and along topo-sequences. Unfortunately, it is mentioned in the paper that ground ice was not specifically taken into account in the analysis due to a lack of data about this aspect.

Some of the results of the modelling makes a lot of sense although other are very surprising. I think the authors should explain better the 'correlations' they obtained. In particular, I would like to see more explanations on the 1) PISR: the peak for ME and the fact that the probability decreases as PISR increase for ALD (is this a ground ice effect? Less PISR, ice closer to the surface?), 2) distance to water (probability decreases and then increases with distance to water for ME and ALD), 3) TWI for ME: probability increases and decreases with rise of TWI.

Again, I would like to stress that I consider the quality of this paper to be good to very good but that the authors would benefit in terms of dissemination and citations to publish it in a different journal with a better-targeted readership.

Note to authors and editors:

• English is not my first language. I provided suggestions to improve the sense of

some sentences, however, I realize that some of them might not be appropriate given my level of English. I therefore leave to the authors some latitude regarding the request for change concerning the English and understand that some of the requested changes won't be implemented. • 'Soil' is used in its geotechnical sense (unconsolidated sediment) and not in terms of pedology.

Specific comments:

Abstract, L12 : The link between high pore water pressure and landscape degradation isn't clear. I understand it but it is implicit in the text. The authors should clarify this in the abstract and later in the text. Perhaps by stressing which geomorphological processes can be triggered by high pore water pressure, how high PWP are generated and how these geomorphological processes can have an impact on landscape evolution, landforms, or, to a different scale, active-layer/surface dynamics.

Abstract, L17-18: 'distance to water' repeated in the same sentence. Correct please.

Abstract, L20: delete 'accurately'. Let the reader judge if this was indeed 'accurately modelled'.

Abstract, L22-23: the authors use the term 'relatively' (. . .low PISR, . . .far from water). I propose to eliminate relatively and suggest to change to something like 'lowest PISR' or simply 'low PISR' and 'far from water' or 'farthest away from water'.

Abstract, L23: '. . . areas that may be sensitive to high PWPs'. This sentence weakens the abstract. I think it is reasonable to say: '. . .areas sensitive to high PWPs' without 'that may be. . .'.

Introduction, L30: delete 'seasonal'. The active layer is a seasonal phenomenon.

Introduction, L31: 'water and ice enrichment at the base of the active layer'. I believe the authors should add 'and in the upper part of permafrost'.

p. 2, L3: 'during the summer months'. This should be either deleted or 'beginning of

winter' be added. Indeed, the bottom of the active layer often thaws as the top of the active layer is refreezing.

p. 2, L4-5: 'During the fall freeze-back period this water undergoes refreezing, consequently developing an ice-rich transient layer at the base of the active layer (Hinkel et al., 2001; Kokelj and Burn, 2003, Shur et al., 2005).'

The transient layer is not explained properly here. The authors have to explain that this water refreezes and remains in the 'permafrost portion' of the soil column during cold year (s) whereas during warm years the transient layer thaws partially, that is the active layer deepens (thawing of the active layer and upper portion of permafrost). The following two years (or more), depending if these years are colder or warmer than the previous ones, the active layer will continue to deepen or the lower portion of it will not thaw and then will be part of the upper permafrost. The authors should re-write the text around the concept of transient layer.

p. 2, L7-8: 'This addition of moisture, as well as infiltration from late season precipitation, results in high pore-water pressures (PWP) at the base of the active layer'. This is the case for saturated (porosity filled with ice and, upon melt, with water) fine-grained soils essentially. Unsaturated sediment will not develop high pore water pressure upon thaw and coarse sediment will usually drain and won't develop high pore water pressure. Please specify. The reference cited could be improved, perhaps cite specific studies concerning pore water pressure in permafrost environment or classic geotechnical literature about PWP and mass movements.

p. 3, L20-22: 'The site is underlain by Devonian sandstone and siltstone bedrock comprising the Weatherall, 20 Hecla Bay, Beverley Inlet and Parry Islands (Burnett Point Member) formations (Harrison, 1995), but outcrops are uncommon'. I suggest to change for: 'The site is underlain by sandstone and siltstone bedrock but outcrops are uncommon (Harrison, 1995).'

p. 4, L24-25: the reason why distance to water (10 m) and distance to ALD (20 m)

[Figure]

needs to be explained. 10 m from water appears close to me for the topic and scale of the study.

p. 5, L4-5: the reason why large spatial clusters of ME were removed from the analysis needs to be explained. It could indeed be interesting to see these large clusters.

p. 6, L5-6: what is the scale of the surficial deposit map used? Could this map along with the marine limit elevation be used to infer, although very generally, the potential distribution of ground ice, given the general relation between grain-size distribution, frost-susceptibility and ground ice? The lack of data on ground ice is, in my view, one of the main weakness of that paper.

p. 6, L10-11: 'While ground ice content is linked to high PWPs, it is not used as an input variable as ground ice maps were unavailable and impractical to attain'. The authors mentioned that ground ice is more abundant below the marine limit (p. 5, L17-19). Was there a factor/weight added to the cells below the marine limit as PWP is more likely to be generated in areas with high ground ice content? Please describe surficial sediment/(cryo) stratigraphy above and below the marine limit. Models indicated 50 and 80 m as key elevations. What's going on around these elevations that could help understand the output of the model better?

p. 11, L20: 'Landscapes composed of fine-grained surficial sediments are susceptible to a wide range of permafrost degradation processes, including the development of high PWP in the active layer'. The development of high PWP is not a permafrost degradation process. High PWP and excess PWP lead to the development of mass movement and this could be included as a 'permafrost degradation process'. Please change.

p. 11, L25-26: 'While soil PWP measurements are not available to confirm pressurization in these instances, the inferred mechanism is diapirisation of sediment slurries from the base of the active layer caused by pore-water pressurization due to ice thaw' Diapirism of sediment slurries can be from the base of the active layer or from lateral

mass movement originating from upslope (there will be mass transfer, at least water, even with low angle slope). I also agree that is it probably more related to the base of the active layer, however the authors haven't shown data to support it. Furthermore, the liquid limit threshold can be attained due to water release upon ice thaw but it can also be attained by the infiltration of rain in the active layer or from subsurface flow. This should be mentioned.

p. 12, L11-13: 'Hence, while surficial materials are broadly similar across CBAWO, the landscape zonation of these two features appears to follow a slope continuum.' I agree and I think the authors should expand their explanation here. Please put this sentence in the context of High-Arctic polar desert watershed/toposequence so that readers could verify if these observations apply in other similar landscape settings. Clarify the link between toposequence, hydrology, moisture and the thermal regime of the active layer.

p. 12, L15-17: 'In 2007, the warmest year since regional records began in 1949, deep active layer development and late July rainfall triggered widespread ALD formation.' I would like to have more information about the effect of rainfall on ALD. There's not enough information about it in the paper, even though it could be an important factor. If rainfall data are available, they should be included in the results and discussed later in the paper.

p. 12, L20-22: 'Similar conditions were observed with MEs associated with terminated active layer fractures in 2012 further suggesting the presence of fluid slurries in situations approaching those that generate ALDs.' …' These observations suggest that MEs, while clearly reflecting evidence for subsurface soil water pressurization also likely play a stabilization role through pressure release to the surface.' 'By contrast, ALDs are associated with sufficient pressurization to induce slope fracturing and downslope movement.' Are the authors suggesting that ME reduced the PWP and reduced therefore the occurrence of ALD? Please make it clear. Is it possible that ME occurred at the location of ALD prior to the slide? I would like the authors to provide

their interpretation/opinion about this point. This can form interesting working hypotheses for future studies.

p. 14, L23-25: 'The susceptibility models demonstrate that ALDs are most probable on hillslopes with gradual to steep slopes and relatively low PISR, whereas MEs are associated with higher elevation areas, low slope angles and in areas relatively far from water (drier)'. I suggest to add concave slope for ALD and convex slope for ME.

Format:

- For all the text: add space between number and unit. Ex: 100 m.

- In the pdf version, at several places, space is missing between words, punctuation, units, etc.

- Figure 1: add scale (1a, d), add complete date (a, b, c, d)

- Figure 6: add scale and complete date. Is the ALD visible in the background or are they more MEs? Please clarify in the figure caption or directly in the figure.

- Table 2. It would be interesting to add some basic statistics to this table. The table provides mean values of terrain variables. Please add the range, the median and the standard deviation for these variables. It would be very useful if one's want to compare this study with other studies conducted in similar/different environmental set-ups.

---

## Author Comment (AC1) · 1 Mar 2017

We appreciate the Reviewer's comments below and we have responded to the points in **bold text**.
* * *
**Reviewer 1:**

General Comments:
This paper employs GIS to define the terrain susceptible to active layer detachment slides and mud ejection features at a High Arctic site. The paper provides a suitable GIS tool to locate these features at Cape Bounty, an approach which may be applicable for similar terrain elsewhere. The paper is quite technical, with lots of jargon so I thought that it might be better suited for a GIS specific journal or PPP-Permafrost and Periglacial Processes where readers might be more knowledgeable about pore water pressure phenomena. If it is to be published in Cryosphere, more effort could be made to relate the research to other cryospheric types/regions where pore water pressure occurs to elucidate processes across diverse icy bodies. Clarification about distance to water, potential incoming solar radiation in the model is required. It was not clear from the paper whether one is concerned with upslope water or downslope water, or water sources in both directions. Also, what defines water (puddle, stream, lake or pond)? It was also not clear the time frame that PISR was calculated for (1 month, 2 months, 2 weeks)?

**Response: Thank you for your comments. However, we disagree about the appropriateness of the journal. My coauthors and I feel that our paper is highly relevant to The Cryosphere as it presents new research on features that are unique to permafrost landscapes. Mud ejections in particular represent a significant gap in the literature. We have made an effort to relate the observations of the features we see at our site to other areas, in particular discussing how pore-water pressure (PWP) results in instability in other regions. We also clarified the model variables (distance to water, PISR, etc.) and believe that after incorporating the comments from the reviewers it will be suitable for publication in TC.**

The study's introduction indicated that rainfall was an important factor in triggering pore water pressure but there was little information about this variable in the study. I think that more effort could also be made to get information about ground ice conditions at the site. I believe that there is a GSC report for this area, from the mid-70's which might report some of this information. Another look at field photos, or data from other studies in this location might provide clearer indication of ice content.

**Response: The introduction has been reworded. Factors impacting PWP are either intrinsic (ex. slope, drainage, solar radiation) or extrinsic (temperature, rainfall) and although extrinsic factors are important, this model only identifies intrinsic factors. Similarly, all areas across the landscape experience relatively homogeneous rainfall, and it is only certain locations which have high PWP, ALDs and MEs due to specific properties of the landscape at these locations. Therefore, we are using this model to identify these landscape variables. This section of the discussion has been removed, and the text has been reworded to clarify this.**

**The ground ice maps you mention don't have sufficient ground ice data for our area or detail. Permafrost cores have been taken at the site near and ALD, and the data shows ice enrichment from 60-80 cm bgs (Lamhonwah et al., in press). Observations in the headwalls of ALDs show ~0.5m of massive ice starting at ~80cm. Additional information has been added to the text.**

Specific Comments:
1) In the abstract you indicate distance to water but is that distance to water upslope of the feature or downslope or both? It is also not clear in the rest of the text.

**Response: We are referring to distance to downslope water sources. Distance to water was calculated using the Euclidean Distance Tool in ArcGIS and distances were measured from a ALD or ME to a hydrological vector layer. Text has been clarified throughout the manuscript to ensure that the difference between distance to water and TWI is clear.**

2) In your abstract perhaps indicate that the GAM model is a GIS-type model.

**Response: The GAM model is not a GIS-type model, it is a statistical model. We used the terrain variables (which were derived using GIS) as inputs into the statistical model.**

3) Be more specific in your abstract about PISR, instead of "relatively low PISR", perhaps put a value in. Is PISR calculated for the whole summer, a few weeks? It is also not clear in your paper. Did you measure solar radiation directly at your study site? If so, how do these values compare with PISR.

**Response: A value has been added for PISR in the text.**

4) In your abstract, perhaps put…Based on these results, this GIS method identifies….
**Response: This has been changed.**

5) At the beginning of the abstract, you indicate that late season precipitation is important for these features to develop but you don't use precipitation as an explanatory variable. In fact, I don't see any information about precipitation or late-season precipitation in the paper.

**Response: The introduction has been reworded. Rainfall is a trigger for high PWP but does not explain sensitivity of the landscape to PWP, so less emphasis was put on rainfall in the introduction.**

6) Again in your abstract, can you be more definite about distance…avoid saying….areas relatively far from water. Again is that upslope or downslope.

**Response: This has been reworded.**

7) Page 2, Line 20-21. Perhaps cut down on the number of references to GAM.

**Response: Some of the older references have been removed.**

8) Lines 27-28. Can you cut down on the references?

**Response: Two of the references which weren't necessary have been removed.**

10) Page 3 Line 10. Put <10 m. There are some other places where you need to leave spaces…see line 14, 27, etc.

**Response: This has been fixed throughout the manuscript.**

11) Line 13. Since you are concerned with the spring/summer period for slope failure, besides the mean annual temperature add information about the spring/summer temperature and also add information about these infrequent, high magnitude precipitation events.

**Response: Mean July temperature has been added to the text, summer precipitation totals, and information regarding the major rainfall events.**

12) Line 25. Provide information on the summer temperature in 2007 and heavy rainfall.

**Response: Mean July temperatures have been added for 2007 and information about the major rainfall events.**

13) Line 27. Again put the temperatures in for 2011 and 2012, and maybe indicate how these temperatures compare with other areas in the High Arctic, and what other scientists were observing (glacial ice loss, sea ice). This will help put your work in context of other cryospheric phenomena.

**Response: Mean July temperature has been added for 2011 and 2012. Above average temperatures were recorded in 2007 and 2012 in other areas of the arctic, and reference has been made to the SWIPA report to put this in context.**

14) Page 4, Line 24. Why did you select >10 m for distance to a water source and again was that upslope or downslope. Also, why did you select a distance to an ALD of > 20 m? Can you plot those randomly ArcGIS points in your map?

**Response: On average the width of channels are Cape Bounty are substantially less than 10 m. To ensure that randomized points were not placed in a stream a rule of >10 m was selected. This refers to the downslope distance to a water source. Again, to ensure that randomized points were not placed within the boundary of existing ALDs a minimum distance of 20 m was selected. Points were generated using the "Random Point" tool in ArcGIS with the additional criteria ( >10 m from a water source and >20 m from an initiation point) limiting to a minimal extent where they could be placed. The location of the random points has been added to Figure 4.**

[Figure]

Disturbance Susceptibility

Very Low (<50th)
Low (>50th)
Moderate (>75th)
High (>90th)
Very High (>95th)

Active Layer Detachment
Mud Ejection
Randomized Undisturbed Samples

15) Page 5, Line 8. Can you plot the randomly generated control points for MEs (78).

**Response: The location of the random points has been added to Figure 4.**

16) Line 15. Do you have a reference to add to after….they all have the potential to contribute to areas having high PWPs?

**Response: The relation of each variable to drainage, soil moisture, and thus PWP were explained individually throughout that section and references were included for each variable.**

17) Line 20. Again, is it distance upslope or downslope? Be specific, in terms of water, is it water table or a creek or a stream or a lake, pond. How do you define water? Many hillslope creeks in the High Arctic dry up after snowmelt, or are intermittent. Do you still estimate distance to them?

**Response: We are referring to distance to downslope water sources. Distance to water was calculated using the Euclidean Distance Tool in ArcGIS and distances were measured from an ALD or ME to a hydrological vector layer which included lakes and rivers (can be seen on Figure 4 and 5). Text has been clarified. These rivers are the larger streams and rivers in the area and remain active throughout the hydrological season.**

18) Line 23. Are you able to compare PISR with measured incoming radiation at a level site to see how they compare over a summer season? If you had a cloudy, rainy season then radiation across the slopes/plateau might not have been critical.

**Response: The mean value for PISR at our site is 1267 MJ/m2, indicating that ALDs have higher probability of occurring where PISR is lower than the site average. More information has been added to put this in context in Section 5.1.**

19) Lines 7-8. Can you say more about the TWI index? How does this compare to the new paradigm of 'spill and fill', which is perhaps a better theory of how water moves in arctic environments (Woo, 2012).

**Response: In this study a FD8 flow algorithm was applied to allow water to flow into multiple neighbouring cells based on the concave or convex nature of the landscape. TWI is an indicator of the likelihood of saturated soil conditions during rain events, and represents hydrologic parameters influenced by slope morphology. TWI provides us with information on where soil moisture is likely to be higher as a result of the accumulation of surface water. This is important as an increase in subsurface water content can lead to increased porewater pressure which is a triggering factor for ALDs and MEs. More detail has been added to the manuscript. Woo (2012) discusses the fill-and-spill concept, and logically this is happening in our area to some extent, however, these subtleties of storage heterogeneity in hillslopes and catchments are difficult to account for using spatially derived data and the landscape scale. However, the TWI index does consider convexity and concavity, and in this manner partitions the slope into various segments.**

20) Lines 10-11. Are you sure that you don't have any information about ground ice content. There must be some geology maps of this area which give an indication of ice content. During your fieldwork, did you not dig a hole in these different landscapes to examine where the ice and moisture were accumulating? Perhaps, look at some of your pictures, particularly, active layer detachment slides. The headwall scarps might give you an indication of where the ice rich depths occur.

**Response: The ground ice maps available for the field area are highly generalized. Permafrost cores have been taken at the site, but the data is unpublished. Observations in the headwalls of ALDs provide further information about ground ice which has been added to the text.**

21) Line 13. Do you really need ρ in front of Sp? Do you have a reference for VIFs?

**Response: The ρ in front of Sp is necessary as this is the notation for this coefficient. The reference Neter et al., 1996 has been added for VIFs.**

22) Page 8. Line 14. Do you have a reference for a confusion matrix?

**Response: There is no reference needed for the confusion matrix as it is a standard methodology (a more complex contingency table).**

23) Lines 20-23. Is this a standard framework for susceptibility/sensitivity? Should you add a reference here?

**Response: It is the dominant method for susceptibility modelling used in the literature. References have been added.**

22) Line 3. In terms of PISR, how does 1100 MJ/m2 compare with what is generally measured during a summer season, and what is the time frame for the PISR estimate (i.e. is this over 30, 60 or 90 days). Do you start your calculations in late August, since you said these features often occur then?

**Response: Total PISR is only calculated for the snow free period which is July 15 – September 15, and this information has been added to the text. The mean value for PISR at our site is 1267 MJ/m2, indicating that ALDs have higher probability of occurring where PISR is lower than the site average. More information has been added to put this in context in Section 5.1.**

23) Line 6. Again are you referring to upslope distance or downslope distance? I would think that upslope distance to water would be more important than downslope.

**Response: Distance to water refers to the downslope distance to a water source and TWI incorporates the upslope contributing area. Downslope distance to water is an indication of drainage and wetness of the landscape, and water sources have the potential to erode banks and cause ALD initiation. This has been added to the text.**

24) Line 16. Indicate the amount of rain which fell late July, also indicate the depth of ground thaw.

**Response: General information about the frequency and magnitude of rainfall has been added throughout the text.**

25) Line 26. What kind of soil structure did you have which allowed these slurries to occur?

**Response: The soils are composed of mineral fines formed in glacial and marine sediments.  We observed desiccation cracking at the site and MEs coming out of cracks in the ground. More information on this has been added.**

---

## Author Comment (AC2) · 1 Mar 2017

We appreciate the Referee's comments below and we have responded to the points in **bold text**.
* * *
**Referee 2:**

The paper uses a general additive model and terrain characteristics derived from remote sensing to map susceptibility of permafrost disturbances (active layer detachment and mud ejection). The GIS-based analysis was successful at identifying important terrain controls at the study site, and the approach seems to have potential for application at other sites. The results are interesting and well executed and the topic is of interest to readers of The Cryosphere, but I'm not convinced The Cryosphere is the most appropriate journal. The paper is quite technical and might be appropriate for a remote sensing journal or for Permafrost and Periglacial Processes, which has a geomorphology focus. An indicator here is that not a single Cryosphere paper was cited. This paper could be made more relevant to The Cryosphere by expanding the discussion to explore consequences for other sites, and by discussing in more depth the physical reasons for the observed explanatory power of the various terrain characteristics.

**Response: We thank the reviewer for their comments, but disagree about the appropriateness of the journal. My coauthors and I feel that our paper is highly relevant to The Cryosphere as it presents new research on features that are unique to permafrost landscapes. Mud ejections in particular represent a significant gap in the literature. We have added information to expand the discussion and think that after incorporating the comments from the reviewers it will be suitable for publication in TC. We have made an effort to relate the observations of the features we see at our site to other areas, in particular discussing how PWP results in instability in other regions. We have added more discussion about the terrain variables to Section 5.1 (see specific comments below).**

Specific comments

The title, abstract, and beginning of the paper focuses on pore water pressure, but the effect of interest is disturbance. High pore-water pressure is not observable directly, and it's possible to have high pore-water pressure without an ALD or ME. The title and the introduction should be revised to better reflect the topic of the paper - susceptibility to disturbance, not pore-water pressure.

**Response: Without high PWP there would be no ALDs or MEs, so we are using the presence of ALDs and MEs as the surface expression of PWP. It is widely accepted in the literature that ALDs and MEs form from high PWP (Washburn, 1956; Shilts, 1978; Zoltai, 1978; French, 2007; Lewkowicz, 2007). We used the presence of ALDs and MEs to predict areas across the landscape that are susceptible to high PWP, and therefore potentially future formation of ALDs and MEs. In this way we feel that the title is appropriate. The introduction has been reworded for clarity.**

Some of the observed relationships between the terrain variables make sense physically and some are counterintuitive. For example, why would ALD be more likely in areas of low PISR? Why would ME's be more likely in drier locations and higher elevations? Physical reasons for

all the observed relationships and especially the counterintuitive ones should be explained to convince the reader that those relationships are real and not spurious correlations.

**Response: Section 5.1 has been elaborated to explain observed relationships between the high susceptibility zones for ALDs and MEs and the terrain variables.**

The probability of observing an ALD approaches 100% for low PISR. This is clearly site-specific and raises concerns about the transferability of the results. Please explain.

**Response: Based on our field mapping, modelling and the subsequent terrain analysis we believe that this research identifies a link between PISR and slope disturbance that can't be ruled out without further examination. While this link may not be a direct relationship, PISR may act as a proxy for additional processes associated with ALD initiation. This is not a new observation as evidence of this relation has been noted at other locations and is cited in the manuscript (Leibman, 1995; Huscroft *et al*., 2004; Lipovsky and Huscroft, 2007; Niu *et al*., 2014). This relation has also been documented at other sites in the High Arctic through the development of susceptibility models (Rudy et al., 2016a and 2016b). More text has been added to section 5.1 discussing the relation of ALDs to low PISR.**

Rainfall is likely to be an important controlling variable. This needs to be discussed, since it is not addressed.

**Response: More specific details have been added about the frequency and magnitude of rainfall events, but rainfall wasn't a variable we looked at in this study. Factors impacting PWP are either intrinsic (ex. slope, drainage, solar radiation) or extrinsic (temperature, rainfall) and although extrinsic factors are important, this model only identifies intrinsic factors. Similarly, all areas across the landscape experience relatively homogeneous rainfall, and it is only certain locations which have high PWP, ALDs and MEs due to specific qualities of the landscape at these locations. Therefore, we are using this model to identify these landscape variables. This section of the discussion has been removed, and the text has been reworded to clarify this.**

Pg 4 Line 24: what's the basis for the constraints >10 m from water source and > 20m from an ALD?

**Response: On average the width of channels at Cape Bounty are less than 10 m, to ensure that randomized points were not placed in a stream a rule of >10 m was selected. Again, to ensure that randomized points were not placed within the boundary of existing ALDs a minimum distance of 20 m was selected. Points were generated using the "Random Point" tool in ArcGIS with the additional criteria ( >10 m from a water source and >20 m from an initiation point). The text has been clarified.**

Pg 5 Line 4-7: The description of the declustering process is difficult to follow and should be explained more clearly. As I understand it, because closely located features carry redundant information, spatial clusters of features are replaced by representative features.

**Response: The reviewer is correct, dense clusters of MEs were removed to avoid redundancy and statistical bias. Declustering was achieved by creating a 10 m buffer zone around each mapped ME feature in ArcGIS, and areas where buffer zones intersected were treated as one large polygon to represent the region of the cluster. A single point representing a ME was randomly generated as a representative point for every 10 clustered points within the polygon (i.e., a cluster of 25 MEs would result in 3 points). This has been reworded in the text for clarification.**

Pg 5. Line 4-7. The declustering algorithm seems arbitrary. Are the results sensitive to how that is done?

**Response: Analysis was done both with and without declustering, and was more representative of the study area with the declustering as it reduced statistical bias to the landscapes where the clusters were found. Clusters of MEs were mapped as a polygon, and then one point for every 10 MEs within the polygon were randomly generated within the area as a representative point.**

Pg 6. Line 21. How were the features partitioned between the calibration and validation subsets? Random?

**Response: The total (combined disturbed and undisturbed points) datasets for each MEs and ALDs were randomly subdivided into 70 per cent calibration and 30 per cent validation subsets. The text has been clarified.**

Pg 26. Line 1. What is an "explained deviance"?

**Response: Relative importance of variables was evaluated by the change in explained deviance from the full model as variables were removed individually. If the variable is important for the model it will result in a higher explained deviance. For example, the slope variable had the greatest explained deviance from the full ALD model. It is the equivalent of $R^2$ in a linear-regression model. This has been clarified and some references have been added.**

Final sentence: The phrase "incentive and potential to move towards: : :" makes for a weak conclusion. Is it not possible to say something more definitive?

**Response: This has been reworded.**

---

## Author Comment (AC3) · 1 Mar 2017

We appreciate the Reviewer's comments below and we have responded to the points in **bold text**.
* * *
**Anonymous Referee #3:**

**General comments:**

In their paper, the authors evaluate the susceptibility of High-Arctic permafrost terrain to disturbances (ALD and ME) related to high pore water pressure. To do so, they used a GIS-based approach, statistics, and field validation, and made the demonstration that such an approach is exportable to other sites. The results indicate that terrain characteristics of ALDs and MEs differed in the modelled high susceptibility zones, whereas they were similar in low susceptibility zones. They have shown that slope was the main variable driving ALD initiation and distance to water was the most important variable explaining ME formation. Although this paper makes an interesting contribution to permafrost landscape hazards and permafrost landscape dynamics studies, I think it would be better suited for a GIS-dedicated journal or a hazards-dedicated journal. Indeed, my impression is that the cryospheric components in this article are not developed sufficiently to justify a publication in Cryosphere. If the editor decides differently, then the authors should develop a section on ground ice and particularly clarify the concept of 'transient layer' and how it applies at the landscape scale, how to model it and how to incorporate it in their GIS-based approach. A point should be made about the distribution of ground ice in a given watershed and along topo-sequences. Unfortunately, it is mentioned in the paper that ground ice was not specifically taken into account in the analysis due to a lack of data about this aspect. Some of the results of the modelling makes a lot of sense although other are very surprising. I think the authors should explain better the 'correlations' they obtained. In particular, I would like to see more explanations on the 1) PISR: the peak for ME and the fact that the probability decreases as PISR increase for ALD (is this a ground ice effect? Less PISR, ice closer to the surface?), 2) distance to water (probability decreases and then increases with distance to water for ME and ALD), 3) TWI for ME: probability increases and decreases with rise of TWI. Again, I would like to stress that I consider the quality of this paper to be good to very good but that the authors would benefit in terms of dissemination and citations to publish it in a different journal with a better-targeted readership.

Response: We thank the anonymous referee for their constructive comments. We have added as much information about ground ice as is available at our study site currently. This includes permafrost cores which have recently been taken at the site near an ALD, and the data shows ice enrichment from 60-80 cm bgs (Lamhonwah et al., in press). Secondly, observations in the headwalls of ALDs show ~0.5 m of massive ice starting at ~80 cm.

**Section 5.1 has been elaborated to explain observed relationships between the high susceptibility zones for ALDs and MEs and the terrain variables.**

**Specific comments:**

Abstract, L12 : The link between high pore water pressure and landscape degradation isn't clear. I understand it but it is implicit in the text. The authors should clarify this in the abstract and later

in the text. Perhaps by stressing which geomorphological processes can be triggered by high pore water pressure, how high PWP are generated and how these geomorphological processes can have an impact on landscape evolution, landforms, or, to a different scale, active-layer/surface dynamics.

**Response: This has been reworded. We have also re-written the introduction to be more clear about how PWP are generated and the impacts of ALDs and MEs on the landscape.**

Abstract, L17-18: 'distance to water' repeated in the same sentence. Correct please.

**Response: This has been corrected.**

Abstract, L20: delete 'accurately'. Let the reader judge if this was indeed 'accurately modelled'.

**Response: This has been deleted.**

Abstract, L22-23: the authors use the term 'relatively' (...low PISR, ...far from water). I propose to eliminate relatively and suggest to change to something like 'lowest PISR' or simply 'low PISR' and 'far from water' or 'farthest away from water'.

**Response: More specific details have been added to this section.**

Abstract, L23: '... areas that may be sensitive to high PWPs'. This sentence weakens the abstract. I think it is reasonable to say: '... areas sensitive to high PWPs' without 'that may be .'.

**Response: This has been changed.**

Introduction, L30: delete 'seasonal'. The active layer is a seasonal phenomenon.

**Response: This has been deleted.**

Introduction, L31: 'water and ice enrichment at the base of the active layer'. I believe the authors should add 'and in the upper part of permafrost'.

**Response: This has been added.**

p. 2, L3: 'during the summer months'. This should be either deleted or 'beginning of winter' be added. Indeed, the bottom of the active layer often thaws as the top of the active layer is refreezing.

**Response: It has been deleted.**

p. 2, L4-5: 'During the fall freeze-back period this water undergoes refreezing, consequently developing an ice-rich transient layer at the base of the active layer (Hinkel et al., 2001; Kokelj and Burn, 2003, Shur et al., 2005).' The transient layer is not explained properly here. The authors have to explain that this water refreezes and remains in the 'permafrost portion' of the

soil column during cold year (s) whereas during warm years the transient layer thaws partially, that is the active layer deepens (thawing of the active layer and upper portion of permafrost). The following two years (or more), depending if these years are colder or warmer than the previous ones, the active layer will continue to deepen or the lower portion of it will not thaw and then will be part of the upper permafrost. The authors should re-write the text around the concept of transient layer.

**Response: We have reworded the text and better explained the idea of the transient layer.**

p. 2, L7-8: 'This addition of moisture, as well as infiltration from late season precipitation, results in high pore-water pressures (PWP) at the base of the active layer'. This is the case for saturated (porosity filled with ice and, upon melt, with water) fine-grained soils essentially. Unsaturated sediment will not develop high pore water pressure upon thaw and coarse sediment will usually drain and won't develop high pore water pressure. Please specify. The reference cited could be improved, perhaps cite specific studies concerning pore water pressure in permafrost environment or classic geotechnical literature about PWP and mass movements.

Response: This is a general statement about how high PWP is generated in areas with icerich transient layers. Previous work in the study site indicates fine-grained soils throughout the area. Similarly an ice-rich layer at the top of the permafrost has been observed at the site from cores (Lamhonwah et al., in press). This information has been added to the study site section and expanded on throughout the text. The references have been updated to include more classic geotechnical literature about slope instability, and this section of the introduction has been rewritten for clarity.

p. 3, L20-22: 'The site is underlain by Devonian sandstone and siltstone bedrock comprising the Weatherall, 20 Hecla Bay, Beverley Inlet and Parry Islands (Burnett Point Member) formations (Harrison, 1995), but outcrops are uncommon'. I suggest to change for: 'The site is underlain by sandstone and siltstone bedrock but outcrops are uncommon (Harrison, 1995).'

**Response: Thank you for the constructive suggestion and this has been changed.**

p. 4, L24-25: the reason why distance to water (10 m) and distance to ALD (20 m) needs to be explained. 10 m from water appears close to me for the topic and scale of the study.

Response: On average the width of channels at Cape Bounty are less than 10 m, to ensure that randomized points were not placed in a stream a rule of >10 m was selected. Again, to ensure that randomized points were not placed within the boundary of existing ALDs a minimum distance of 20 m was selected. Points were generated using the "Random Point" tool in ArcGIS with the additional criteria (>10 m from a water source and >20 m from an initiation point). The text has been clarified.

p. 5, L4-5: the reason why large spatial clusters of ME were removed from the analysis needs to be explained. It could indeed be interesting to see these large clusters.

Response: Analysis was done with and without declustering, and was more representative of the study area with the declustering as it reduced statistical bias to the landscapes where the clusters were found. Declustering was achieved by creating a 10 m buffer zone around each mapped ME feature in ArcGIS 10.1, and areas where buffer zones intersected were treated as one large polygon to represent the region of the cluster. A single point representing a ME was randomly generated as a representative point for every 10 clustered points within the polygon (i.e., a cluster of 25 MEs would result in 3 points). This has been reworded in the text for clarification.

p. 6, L5-6: what is the scale of the surficial deposit map used? Could this map along with the marine limit elevation be used to infer, although very generally, the potential distribution of ground ice, given the general relation between grain-size distribution, frost-susceptibility and ground ice? The lack of data on ground ice is, in my view, one of the main weakness of that paper.

Response: We did not use a surficial deposit map for this analysis, and such a map does not exist for this site. We did use marine limit (elevation) as a proxy for ground ice, as generally there will be finer-grained sediment below marine limit and thus more ground ice. This is stated in section 3.3 of the methods. We've added more data on the ground ice conditions at the site.

p. 6, L10-11: 'While ground ice content is linked to high PWPs, it is not used as an input variable as ground ice maps were unavailable and impractical to attain'. The authors mentioned that ground ice is more abundant below the marine limit (p. 5, L17-19). Was there a factor/weight added to the cells below the marine limit as PWP is more likely to be generated in areas with high ground ice content? Please describe surficial sediment/(cryo) stratigraphy above and below the marine limit. Models indicated 50 and 80 m as key elevations. What's going on around these elevations that could help understand the output of the model better?

Response: The estimation of marine limit at the site is approximately 60-80 m, but it is a diffuse gradient that is not clearly defined, so we didn't put any weight on the cells below marine limit. We do not have data for ground ice conditions above and below marine limit, and the surficial sediments do not show a clear difference above and below marine limit.

Note that table 2 has been updated and 60m is a key elevation for MEs, attributed to drier, barren, plateau environments which have deeper annual thaw. Results indicate that 50 m is a key elevation for ALDs, which is below the marine limit of 60-80 m for the site indicating that ground ice likely plays a role here. More information has been added to section 5.1 on this matter.

p. 11, L20: 'Landscapes composed of fine-grained surficial sediments are susceptible to a wide range of permafrost degradation processes, including the development of high PWP in the active layer'. The development of high PWP is not a permafrost degradation process. High PWP and excess PWP lead to the development of mass movement and this could be included as a 'permafrost degradation process'. Please change.

**Response: This has been reworded.**

p. 11, L25-26: 'While soil PWP measurements are not available to confirm pressurization in these instances, the inferred mechanism is diapirisation of sediment slurries from the base of the active layer caused by pore-water pressurization due to ice thaw' Diapirism of sediment slurries can be from the base of the active layer or from lateral mass movement originating from upslope (there will be mass transfer, at least water, even with low angle slope). I also agree that is it probably more related to the base of the active layer, however the authors haven't shown data to support it. Furthermore, the liquid limit threshold can be attained due to water release upon ice thaw but it can also be attained by the infiltration of rain in the active layer or from subsurface flow. This should be mentioned.

**Response: Holloway et al. (2016) show evidence for MEs originating at the base of the active layer, and we have referenced this work. MEs mainly occur on flat terrain, so upslope contributions would be limited. There is very limited literature on MEs. We have removed discussion of liquid limits and have clarified the text.**

p. 12, L11-13: 'Hence, while surficial materials are broadly similar across CBAWO, the landscape zonation of these two features appears to follow a slope continuum.' I agree and I think the authors should expand their explanation here. Please put this sentence in the context of High-Arctic polar desert watershed/toposequence so that readers could verify if these observations apply in other similar landscape settings. Clarify the link between toposequence, hydrology, moisture and the thermal regime of the active layer.

**Response: Text has been added to describe the toposequence at our site. We've added information about the hydrology and active layer in the zones of high susceptibility in Section 5.1.**

p. 12, L15-17: 'In 2007, the warmest year since regional records began in 1949, deep active layer development and late July rainfall triggered widespread ALD formation.' I would like to have more information about the effect of rainfall on ALD. There's not enough information about it in the paper, even though it could be an important factor. If rainfall data are available, they should be included in the results and discussed later in the paper.

Response: Response: More specific details have been added about the frequency and magnitude of rainfall events, but rainfall wasn't a variable we looked at in this study. Factors impacting PWP are either intrinsic (ex. slope, drainage, solar radiation) or extrinsic (temperature, rainfall) and although extrinsic factors are important, this model only identifies intrinsic factors. Similarly, all areas across the landscape experience relatively homogeneous rainfall, and it is only certain locations which have high PWP, ALDs and MEs due to specific qualities of the landscape at these locations. Therefore, we are using this model to identify these landscape variables. This section of the discussion has been removed, and the text has been reworded to clarify this.

p. 12, L20-22: 'Similar conditions were observed with MEs associated with terminated active layer fractures in 2012 further suggesting the presence of fluid slurries in situations approaching

those that generate ALDs.'...' These observations suggest that MEs, while clearly reflecting evidence for subsurface soil water pressurization also likely play a stabilization role through pressure release to the surface.' 'By contrast, ALDs are associated with sufficient pressurization to induce slope fracturing and downslope movement.' Are the authors suggesting that ME reduced the PWP and reduced therefore the occurrence of ALD? Please make it clear. Is it possible that ME occurred at the location of ALD prior to the slide? I would like the authors to provide their interpretation/opinion about this point. This can form interesting working hypotheses for future studies.

**Response: We go into further detail in the subsequent section 5.2 about MEs possibly releasing pressures and stabilizing slopes. The text here has been reworded for clarity.**

p. 14, L23-25: 'The susceptibility models demonstrate that ALDs are most probable on hillslopes with gradual to steep slopes and relatively low PISR, whereas MEs are associated with higher elevation areas, low slope angles and in areas relatively far from water (drier)'. I suggest to add concave slope for ALD and convex slope for ME.

**Response: This has been added.**

Format: For all the text: add space between number and unit. Ex: 100 m.

**Response: This has been corrected.**

In the pdf version, at several places, space is missing between words, punctuation, units, etc.

**Response: This has been corrected.**

Figure 1: add scale (1a, d), add complete date (a, b, c, d).

**Response:** Scale has been added to Figure 1 a and d, and complete dates have been added to the figure caption.**

Figure 1: (a) Active layer detachment at Cape Bounty Arctic Watershed Observatory (CBAWO), Melville Island, NU, 28 July 2007. Clay slurry is evident along scar track immediately post disturbance. (b) Active mud ejection occurring on a plateau, 13 July 2012. (c) Clay slurry pooling in a crack at the headwall of a recently initiated active layer detachment, 16 July 2012. (d) Field of inactive MEs on a plateau, 18 June 2012.

Figure 6: add scale and complete date. Is the ALD visible in the background or are they more MEs? Please clarify in the figure caption or directly in the figure.

Response: Scale and complete date have been added. The picture is taken at the edge of the ALD looking out towards the adjacent terrain (the ALD is not visible). This has been clarified.

---

## Author Comment (AC4) · 1 Mar 2017

We thank the reviewers for their comments, and we have responded to each. Find our responses in the supplementary documents.